# Genomic characterization of genes encoding histone acetylation modulator proteins identifies therapeutic targets for cancer treatment

Zhongyi Hu[1], Junzhi Zhou[1], Junjie Jiang[1], Jiao Yuan[1], Youyou Zhang[1,2], Xuepeng Wei[3], Nicki Loo[1], Yueying Wang[1], Yutian Pan[1], Tianli Zhang[1], Xiaomin Zhong[4], Meixiao Long[5], Kathleen T. Montone[6], Janos L. Tanyi[2], Yi Fan[7], Tian-Li Wang[8,9], Ie-Ming Shih[8,9], Xiaowen Hu[1,2] & Lin Zhang[1,2]

A growing emphasis in anticancer drug discovery efforts has been on targeting histone acetylation modulators. Here we comprehensively analyze the genomic alterations of the genes encoding histone acetylation modulator proteins (HAMPs) in the Cancer Genome Atlas cohort and observe that HAMPs have a high frequency of focal copy number alterations and recurrent mutations, whereas transcript fusions of HAMPs are relatively rare genomic events in common adult cancers. Collectively, 86.3% (63/73) of HAMPs have recurrent alterations in at least 1 cancer type and 16 HAMPs, including 9 understudied HAMPs, are identified as putative therapeutic targets across multiple cancer types. For example, the recurrent focal amplification of *BRD9* is observed in 9 cancer types and genetic depletion of *BRD9* inhibits tumor growth. Our systematic genomic analysis of HAMPs across a large-scale cancer specimen cohort may facilitate the identification and prioritization of potential drug targets and selection of suitable patients for precision treatment.

[1] Center for Research on Reproduction and Women's Health, University of Pennsylvania, Philadelphia, Pennsylvania 19104, USA. [2] Department of Obstetrics and Gynecology, University of Pennsylvania, Philadelphia, Pennsylvania 19104, USA. [3] Department of Biochemistry and Biophysics, University of Pennsylvania, Philadelphia, Pennsylvania 19104, USA. [4] Center for Stem Cell Biology and Tissue Engineering, Department of Biology, Zhongshan School of Medicine, Sun Yat-Sen University, Guangzhou 510080, China. [5] Division of Hematology, Department of Internal Medicine, Ohio State University, Columbus 43210 Ohio, USA. [6] Department of Pathology and Laboratory Medicine, University of Pennsylvania, Philadelphia, Pennsylvania 19104, USA. [7] Department of Radiation Oncology, University of Pennsylvania, Philadelphia, Pennsylvania 19104, USA. [8] Departments of Gynecology and Obstetrics, Johns Hopkins University School of Medicine, Baltimore, Maryland 21231, USA. [9] Department of Pathology, Johns Hopkins University School of Medicine, Baltimore, Maryland 21231, USA. These authors contributed equally: Zhongyi Hu, Junzhi Zhou, Junjie Jiang. Correspondence and requests for materials should be addressed to X.H. (email: xiaowenh@pennmedicine.upenn.edu) or to L.Z. (email: linzhang@pennmedicine.upenn.edu)

Histone acetylation modulator proteins (HAMPs), the primary protein families that mediate the modification and recognition of histone acetylation, include histone acetyltransferases (HATs; writers), histone deacetylases (HDACs; erasers), and proteins containing bromodomains (BRD-containing proteins or acetyl-Lys-binding proteins; readers)[1–5]. HATs acetylate the conserved lysine side chains of histone proteins by transferring an acetyl group from acetyl-coenzyme A, thereby forming N-ε-acetyl-L-lysine. In general, chromatin adopts a more relaxed structure after histone acetylation, which enables the recruitment of the transcriptional machinery and increases gene transcription. In contrast, HDACs remove acetyl groups from the N-ε-acetyl-L-lysines of histones, which allows the histones to wrap the DNA more tightly. A BRD is a protein domain (~110 amino acids) that recognizes acetylated lysine residues in histone tails. This recognition is a prerequisite for protein–histone association and chromatin remodeling. An increasing number of HDAC inhibitors (HDACis) have been approved for the clinical care of patients with hematological malignancies and BRD-protein inhibitors are emerging as a new class of anticancer agents that have shown promising therapeutic potential in early clinical trials[5–12].

Recent genomic studies have shown that the genes involved in epigenetic regulation are altered in cancers at unexpectedly high frequencies[13–17], suggesting that certain HAMPs may serve as driver genes during cancer development. Advances in large-scale and multi-dimensional profiles of cancer genomes, such as the Cancer Genome Atlas (TCGA) project, have provided novel resources for identifying potential cancer driver genes and therapeutic targets. In this study, by integrating multi-omic profiles, we comprehensively characterized the genomic alterations of 73 HAMP genes (Supplementary Data 1)[1], including 18 HAT genes, 43 BRD-containing genes (including 6 HATs that also contain BRDs), and 18 HDAC genes, across the whole TCGA data cohort ($n > 10{,}000$, including samples of 33 cancer types from 27 primary sites, Supplementary Data 2). A publicly accessible database was developed to assist researchers with analyzing and visualizing HAMP genomic alteration data through the Functional Cancer Genome data portal (FCG data portal: http://52.25.87.215/home/). Our integrated genomic study indicates that many uncharacterized HAMPs are putative cancer-causing genes with therapeutic potential in certain cancer types.

## Results

### Ubiquitous mRNA expression of HAMPs across cancers.

To characterize the messenger RNA expression of HAMPs in cancer, we analyzed the RNA sequencing (RNA-seq) profiles in TCGA (Supplementary Data 2). We found that most HAMPs were ubiquitously expressed across the 33 cancer types (Fig. 1a and Supplementary Data 3). Only TAF1L was not detected in any of the cancer types examined and BRDT, CECR2, SP140, HDAC9, SIRT4, SP110, and TRIM66 had restricted expression in 5, 17, 23, 25, 30, 32, and 32 cancer types, respectively (Fig. 1b). Similar expressional patterns were also observed in corresponding normal adjacent tissues as well as established cancer cell lines (Supplementary Figure 1). Although these lineage-specific HAMPs were mainly detected in the cancer types derived from the tissues in which the corresponding HAMPs are normally expressed, they were also ectopically expressed in a small fraction of other cancer types. For example, the testis-specific BRD gene BRDT was not solely detected in testicular germ cell tumors (TGCT); it was also found in a small fraction of lung cancers (25.34% of lung adenocarcinomas [LUAD] and 16.97% of lung squamous cell carcinoma [LUSC]), uterine carcinosarcoma (UCS;16.07%), and esophageal carcinoma (ESCA; 11.18%). This finding indicates the therapeutic potential of targeting lineage-specific HAMPs in certain cancer types. Among the ubiquitously expressed HAMPs, TRM28, BRD2, and HDAC1 had remarkably higher mRNA expression levels than the other HAMPs. Unexpectedly, although HAMPs were ubiquitously expressed in cancers, their mRNA expression levels were informative and facilitated the differentiation of the tumor specimens from different cancer types via a machine learning algorithm (t-distributed stochastic neighbor embedding)[18] (Fig. 1c). Notably, the cancer types with related tissue origins clustered together. For example, cancers derived from the digestive tract epithelium (head and neck squamous cell carcinoma [HNSC], ESCA, stomach adenocarcinoma [STAD], colon adenocarcinoma [COAD], and rectum adenocarcinoma [READ]) shared similar HAMP expression signatures. In contrast, melanoma (skin cutaneous melanoma [SKCM] and uveal melanoma [UVM]), hematologic (lymphoid neoplasm diffuse large B-cell lymphoma [DLBC] and acute myeloid leukemia [LAML]), neurological (glioblastoma multiforme [GBM] and brain lower-grade glioma [LGG]), and germline (TGCT) cancers were clearly distinct from the cancers with epithelial origins. Collectively, most HAMPs were ubiquitously expressed across cancer types, but their expression patterns were largely cancer type- and tissue lineage-specific.

### Somatic copy number alterations of HAMPs across cancers.

To characterize the somatic copy number alterations (SCNAs) of HAMPs in cancer, we analyzed the single-nucleotide polymorphism (SNP) array profiles from TCGA (Supplementary Data 2). The putative cancer-causing HAMPs driven by SCNAs in each cancer type were identified using four criteria (Fig. 2a). We initially identified 496 recurrent focal SCNA events in HAMPs across 33 cancer types, and 68 of 73 (93.15%) HAMPs were observed in a significantly recurrent focal SCNA region in at least one cancer type (Supplementary Data 4). This finding is consistent with the recent report that the recurrent SCNA regions in cancer are significantly enriched for genes involved in epigenetic regulation[19]. We next estimated the G-score (Supplementary Data 5), which considers both the amplitude of the aberration and the frequency of its occurrence across samples, for each recurrent focal SNCA event by GISTIC2[20]. The events with low frequency and amplitude were removed (G-score < 0.1). After we removed the non-detectable HAMPs, we analyzed the correlations between mRNA expression and predicted copy number by Pearson's test. A significant and positive correlation was observed for 80.2% (154/192) of HAMP SCNA events identified in the first three steps, strongly suggesting that SCNA is an important mechanism that leads to the dysregulation of HAMPs in cancers.

Across 33 cancer types, we identified 54 HAMPs that met all four criteria in at least one cancer type (Fig. 2b, Supplementary Figure 2, and Supplementary Data 6). For example, the well-known oncogenic HAMP BRD4 was recurrently amplified in six cancer types, including adrenocortical carcinoma (ACC), breast invasive carcinoma (BRCA), ESCA, liver hepatocellular carcinoma, ovarian serous cystadenocarcinoma (OV), and uterine corpus endometrial carcinoma (UCEC) (Fig. 2b). Notably, SCNAs of HAMPs were largely cancer type-specific (Fig. 2b and Supplementary Figure 2): 21 of 54 (38.89%) HAMPs with recurrent SCNAs were only observed in one cancer type and no HAMP SCNA was found in more than nine cancer types. Bladder urothelial carcinoma (BLCA, $n = 16$), sarcoma (SARC, $n = 13$), and LUSC ($n = 12$) had the largest numbers of recurrent SCNAs in HAMPs, whereas DLBC, kidney chromophobe (KICH), kidney renal papillary cell carcinoma (KIRP), pancreatic adenocarcinoma (PAAD), pheochromocytoma and paraganglioma (PCPG), thyroid carcinoma (THCA), and thymoma (THYM) had none

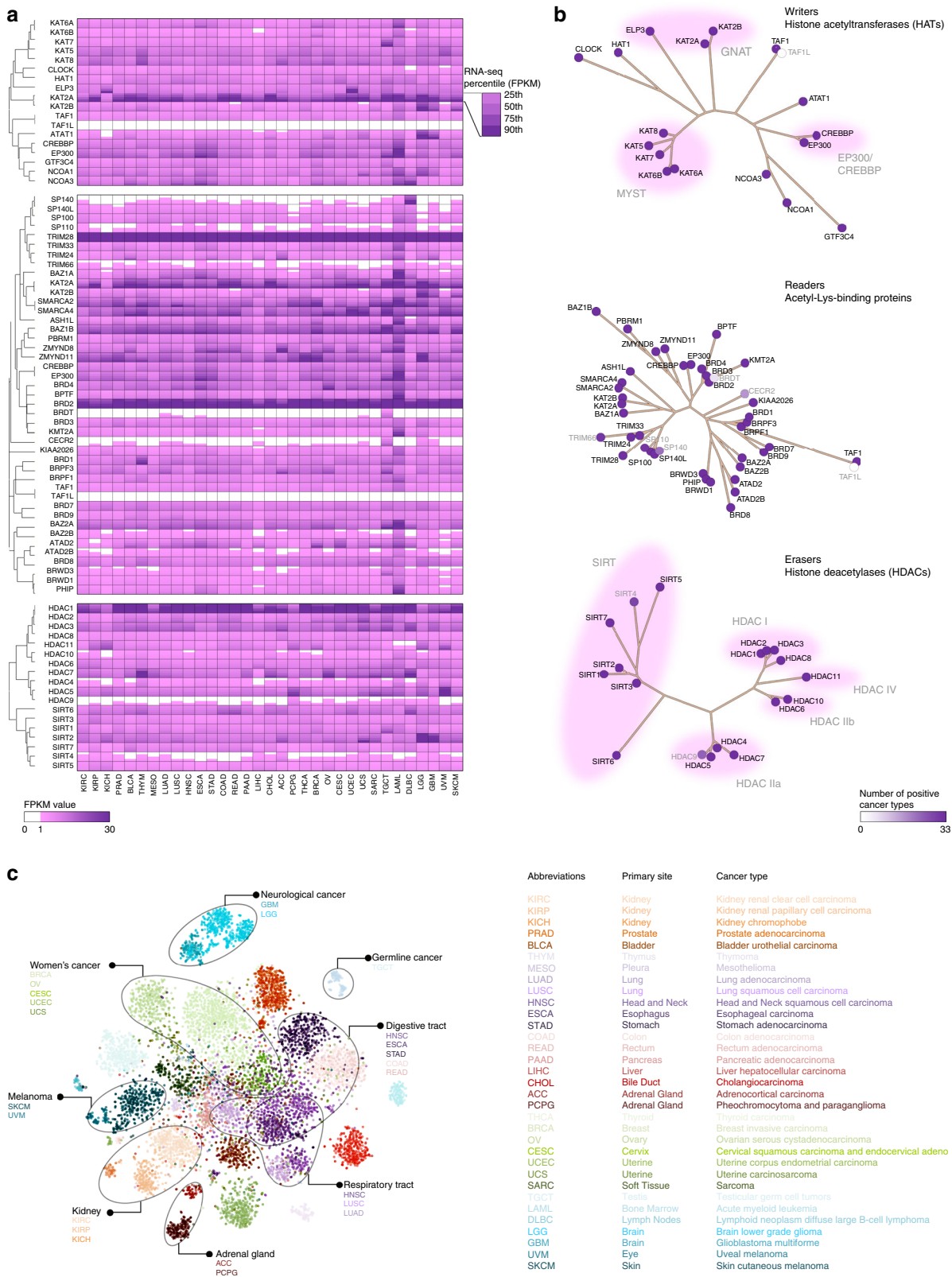

(Fig. 2b and Supplementary Figure 2). Importantly, only six HAMPs had both recurrent copy number gain and loss in different types of tumors. Most HAMPs were consistently altered (either gain or loss) across different cancer types, suggesting that these HAMP SCNAs are not merely the result of genomic instability but may functionally contribute to tumor development. Notably, most identified focal regions contained multiple genes

that co-altered with HAMPs (Supplementary Data 7). In addition, several HAMPs (*SP140/SP140L/SP100/SP110*, *HADC10/BRD1*, *HDAC8/TAF1*, and *NCOA3/ZMYND8*) were co-altered due to their genomic proximity. Among the three types of HAMPs, BRD proteins showed the highest frequency of recurrent SCNAs, whereas HATs presented few recurrent events (Fig. 2c). Five HDAC genes (*HDAC2*, *HDAC4*, *HDAC5*, *HDAC10*, and *SIRT3*)

**Fig. 1** Ubiquitous HAMP mRNA expression across cancers. **a** The heatmap shows the mRNA expression levels of HAMPs across cancers. The intensity of purple indicates the percentile (25th, 50th, 75th, and 90th) of the FPKM value of each HAMP in a given cancer type. The phylogenetic trees were generated by multiple sequence alignments of the full-length sequences of the proteins. **b** Summary of the overall mRNA expression of each HAMP across 33 cancer types. The intensity of purple indicates the number of cancer types in which a given HAMP was defined as detectable (the 90th FPKM value ≥ 1). Gray indicates the HAMPs with mRNA expression levels that are cancer type-specific. The phylogenetic trees were generated by multiple sequence alignments of the full-length sequences of the proteins. **c** TCGA specimens were arranged in two dimensions based on the similarity of their HAMP expression profiles by the dimensionality reduction analysis t-SNE (t-distributed stochastic neighbor embedding). Colors represent cancer types. The cancer types (n = 33) and their color keys are listed based on tissue origin

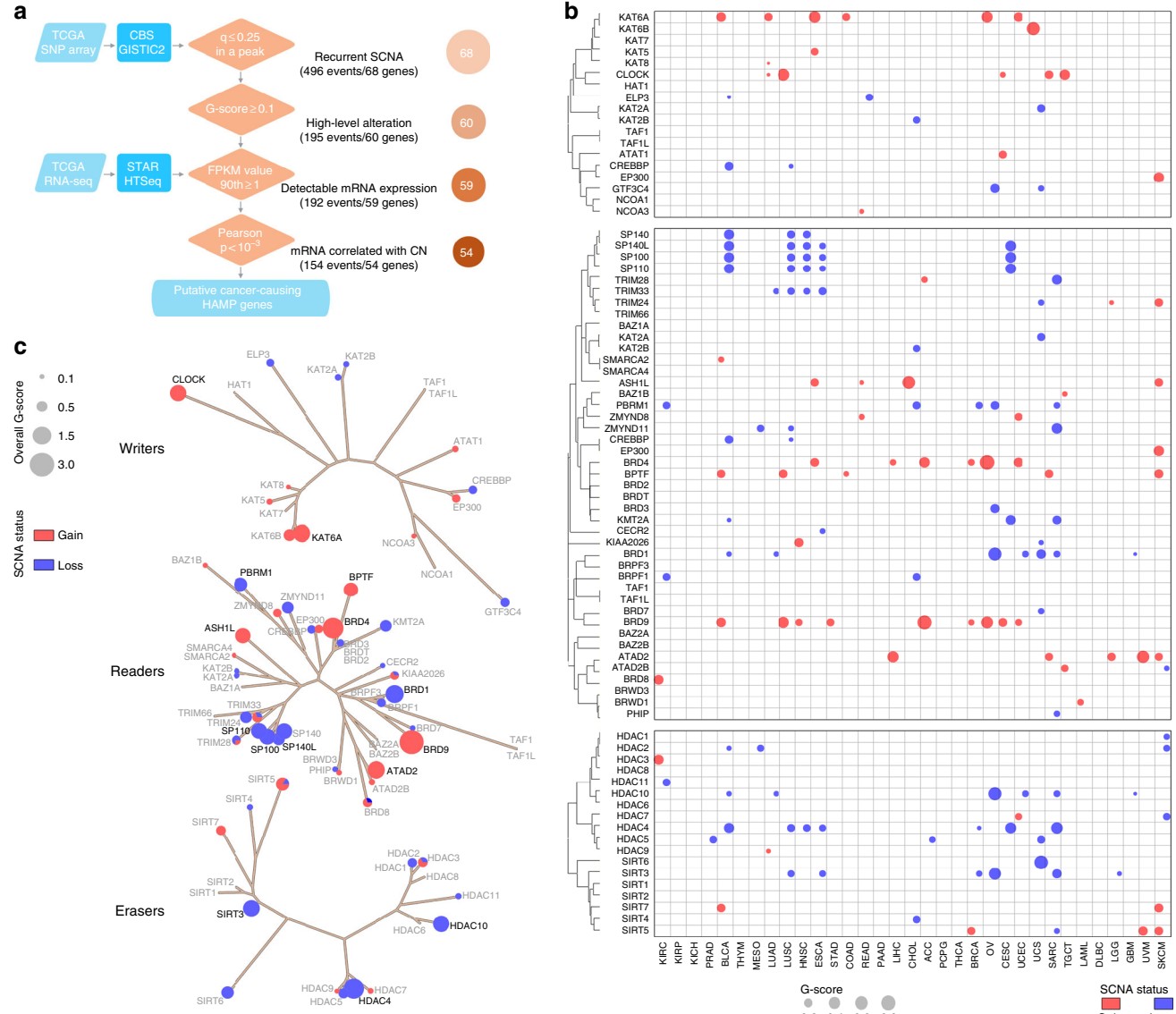

**Fig. 2** Somatic copy number alterations of HAMPs across cancers. **a** The workflow of somatic copy number alteration analysis. Four criteria were used to identify the putative cancer-causing HAMPs driven by SCNAs in each cancer type. The numbers of HAMPs that passed each filter in at least one cancer type are shown on the right. **b** The bubble plot shows the G-scores, which consider both the amplitudes of the aberrations and the frequencies of their occurrence across samples, of the putative cancer-causing HAMPs driven by SCNAs in each cancer type. The size of the bubble: G-score; red: gain; blue: loss. The phylogenetic trees were generated by multiple sequence alignments of the full-length sequences of the proteins. **c** Summary of the overall G-scores of the putative cancer-causing HAMPs driven by SCNAs, which were identified in at least one cancer type. The size of the bubble: the overall G-score; red: gain; blue: loss. The HAMPs coding in black and gray indicate a gene with an overall G-score ≥ 0.8 and < 0.8, respectively. The phylogenetic trees were generated by multiple sequence alignments of the full-length sequences of the proteins

showed copy number loss in at least two cancer types (Fig. 2c). The *HDAC4*, *HDAC10*, and *SIRT3* copy numbers were recurrently lost in eight, six, and six cancer types, respectively (Fig. 2b).

Collectively, we identified 33 HAMPs that recurrently gained or lost copy numbers in more than one cancer type (Fig. 2b). To estimate the SCNAs for these genes at a pan-cancer level, we calculated an overall G-score by an unweighted numeric sum of

G-scores estimated for each cancer type (Fig. 2c and Supplementary Data 8). *BRD9*, *BRD4*, *KAT6A*, *ATAD2*, *CLOCK*, *ASH1L*, and *BPTF* showed high overall G-scores for copy number gain, whereas *HDAC4*, *BRD1*, *SIRT3*, *HDAC10*, *SP100*, *SP140L*, *SP110*, and *PBRM1* had high overall G-scores for copy number loss (overall G-score > 0.9; Fig. 3a–c). Notably, ~3.2–18.8% of HAMP gains appeared to be high-level alterations (GISTIC status = 2; Fig. 3b); in contrast, high-level alterations (GISTIC status = − 2) were markedly less frequent in HAMPs with copy number loss (0.8–9.4%; Fig. 3c). Consistent with this observation, homozygous deletions, as estimated with the ABSOLUTE algorithm, were rare events for these genes. This finding suggests that HAMPs may have critical roles in cell survival and complete loss may be lethal. Taken together, we identified 54 putative cancer-causing HAMPs driven by SCNAs in certain cancer types.

**Somatic mutations of HAMPs across cancers**. To characterize the somatic mutations (single-nucleotide variants and indels) of HAMPs in cancer, we analyzed whole-exome sequencing profiles from TCGA (Supplementary Data 2). The mutation call set was generated via an ensemble calling strategy by the MC3 (Multi-Center Mutation Calling in Multiple Cancers[21]) project, then we integrated five complementary methods to identify the genes with mutations that have significant signs of positive selection during tumor evolution (Fig. 4a). Collectively, across 33 cancer types, we identified 34 HAMPs that have recurrent mutations in at least one cancer type (Fig. 4b, c and Supplementary Data 9 and 10). Although *EP300* and *CREBBP* were widely mutated in multiple cancer types (7 and 4 cancer types, respectively; Fig. 3b), the recurrent mutations of HAMPs were largely cancer type-specific (Fig. 4b): 17 of 34 (50%) HAMPs with recurrent mutations were only observed in one cancer type and no recurrent mutation of HAMPs was found in more than 7 cancer types. Interestingly, except for *PBRM1*, which showed a markedly high mutation frequency in kidney renal clear cell carcinoma (KIRC; 40.1%), most HAMPs showed mutation frequencies < 5% in a given cancer type (Supplementary Data 11). UCEC (n = 20), SKCM (n = 8), BLCA (n = 6), cervical squamous cell carcinoma, and endocervical adenocarcinoma (CESC; n = 6), and HNSC (n = 6) had the largest numbers of recurrent mutations in HAMPs, whereas ACC, DLBC, ESCA, KICH, LAML, LGG, mesothelioma (MESO), OV, PAAD, PCPG, READ, SARC, TGCT, THCA, THYM, UCS, and UVM had none (Fig. 4b and Supplementary Figure 3). Among the three types of HAMPs, BRD proteins and HATs had the highest frequencies of recurrent mutations and HDACs had few recurrent events (Fig. 4c). To estimate the overall recurrent mutations of HAMPs at a pan-cancer level, an overall M-score was calculated by an unweighted numeric sum of M-scores estimated for each cancer type (Fig. 4c and Supplementary Data 12).

Among eight HAMPs with M-scores > 0.4, *EP300*, *PBRM1*, and *CREBBP* had the largest overall M-scores across the 33 cancer types (Fig. 5a). *CREBBP* and *EP300* appeared to be widely mutated in multiple cancer types at a low to modest frequency (0.2–13.5% and 0.2–14.8%, respectively) and *PBRM1* was highly mutated in KIRC (40.1%). At a pan-cancer level, except for *PBRM1*, the most common mutation category of HAMPs was missense mutation (53.0–77.5%; Fig. 5b and Supplementary Data 13) and the dominant mutation type was heterozygous mutations (67.2–89.0%; Fig. 5c and Supplementary Data 14). In contrast, *PBRM1* was most commonly affected by truncating mutations (49.1%; Fig. 5b) and homozygous mutations (37.5%; Fig. 5c). Using the ABSOLUTE algorithm[22], we also determined the timing of the mutational processes and the clonal statuses of the mutations in HAMPs. More than 50% of mutations in

HAMPs were early genomic events (Fig. 5d and Supplementary Data 15) and more than 65% of mutations in HAMPs were clonal mutations (Fig. 5e and Supplementary Data 16), which suggests that targeting these early, clonal mutations may be a novel strategy for the treatment of cancer. We also analyzed the distributions of the mutations across the gene bodies and found that mutations in HAMPs were widely spatially distributed along the entire coding sequences, not concentrated within a specific local region (Fig. 5f). Next, we performed this analysis in each cancer type (Fig. 5g–k and Supplementary Data 13–16). Except for *PBRM1*, most recurrently mutated HAMPs did not show a cancer type-specific pattern for mutation category, type, timing, or clonal status, although BRCA, COAD, GBM, SKCM, STAD, and UCEC had higher mutation frequencies than average. Notably, we observed that *PBRM1* mutations showed a unique pattern in KIRC and cholangiocarcinoma (CHOL). Compared with other cancer types, KIRC and CHOL showed higher frequencies of *PBRM1* mutations (40.1% in KIRC and 19.4% in CHOL vs. an average of 2.3% among other cancer types). Importantly, the dominant *PBRM1* mutation category and type in KIRC and CHOL were truncating (83.9% and 85.7%, respectively) and homozygous (85.2% and 71.4%, respectively) mutations, which were remarkably higher than the averages in other cancer types (30.3% and 19.4%, respectively).

Interestingly, we observed that, in both the *CREBBP* and *EP300* genes, the most frequent mutations were located within the catalytic domain, although the mutations were not statistically significant hotspot mutations at the individual gene level based on OncodriveCLUST analysis (Fig. 5f). We expanded the mutation hotspot analysis from a single gene to a gene family that shares evolutionarily conserved protein modules. To this end, we performed domain mutation analysis using the LowMACA algorithm[23] on five domains that are shared by HAMPs. Seven hotspot positions (p < 0.05, false discovery rate < 0.05, LowMACA algorithm) were identified in the HAT_KAT11, MOZ_SAS, BRD, and Hist_deacetyl domains of HAMPs (Fig. 5l–n and Supplementary Data 17). Consistent with our analysis, one of the hotspot mutations that we identified in the HAT_KAT11 domain was also recently reported in a pan-cancer analysis of protein domain mutations in 5496 tumors[24]. Taken together, we identified 34 HAMPs that may be putative cancer-causing HAMPs driven by somatic mutations in given cancer types.

**Transcript fusions of HAMPs across cancers**. To characterize transcript fusions of HAMPs in cancer, we retrieved the gene fusion data of TCGA from the TumorFusions database[25]. We observed 400 fusion transcripts (349 fusion pairs) of 73 HAMPs in 9799 tumor specimens across 33 cancer types (Fig. 6a and Supplementary Data 18), which suggests that transcript fusion is a rare genetic alteration compared with SCNAs and mutations in HAMPs in common adult cancers. Across all cancer types, UCS (9/57 tumor specimens with fusion information), UCEC (26/180), and LUAD (41/516) had the highest frequencies of HAMP transcript fusion events, whereas only one HAMP fusion was detected in CHOL, COAD, DLBC, TGCT, THCA, and THYM (Supplementary Figure 4 and Supplementary Data 19). Among 400 detected HAMP fusion transcripts, 298 (74.5%), 50 (12.5%), 30 (7.5%), and 22 (5.5%) events were tier 1, tier 2, tier 3, and tier 4, respectively (Fig. 6a and Supplementary Data 20). Only 81 of 400 (20.25%) HAMP fusion transcripts, representing 30 of 349 fusion pairs, were recurrent fusions that occurred at least twice across all cancer types. *KAT6B-ADK* (n = 6), *BPTF-PITPNC1* (n = 5), and *NCOA3-EYA2* (n = 5) were the most frequent fusions among the common cancer types examined in our study (Fig. 6a and Supplementary Data 18). When we analyzed the

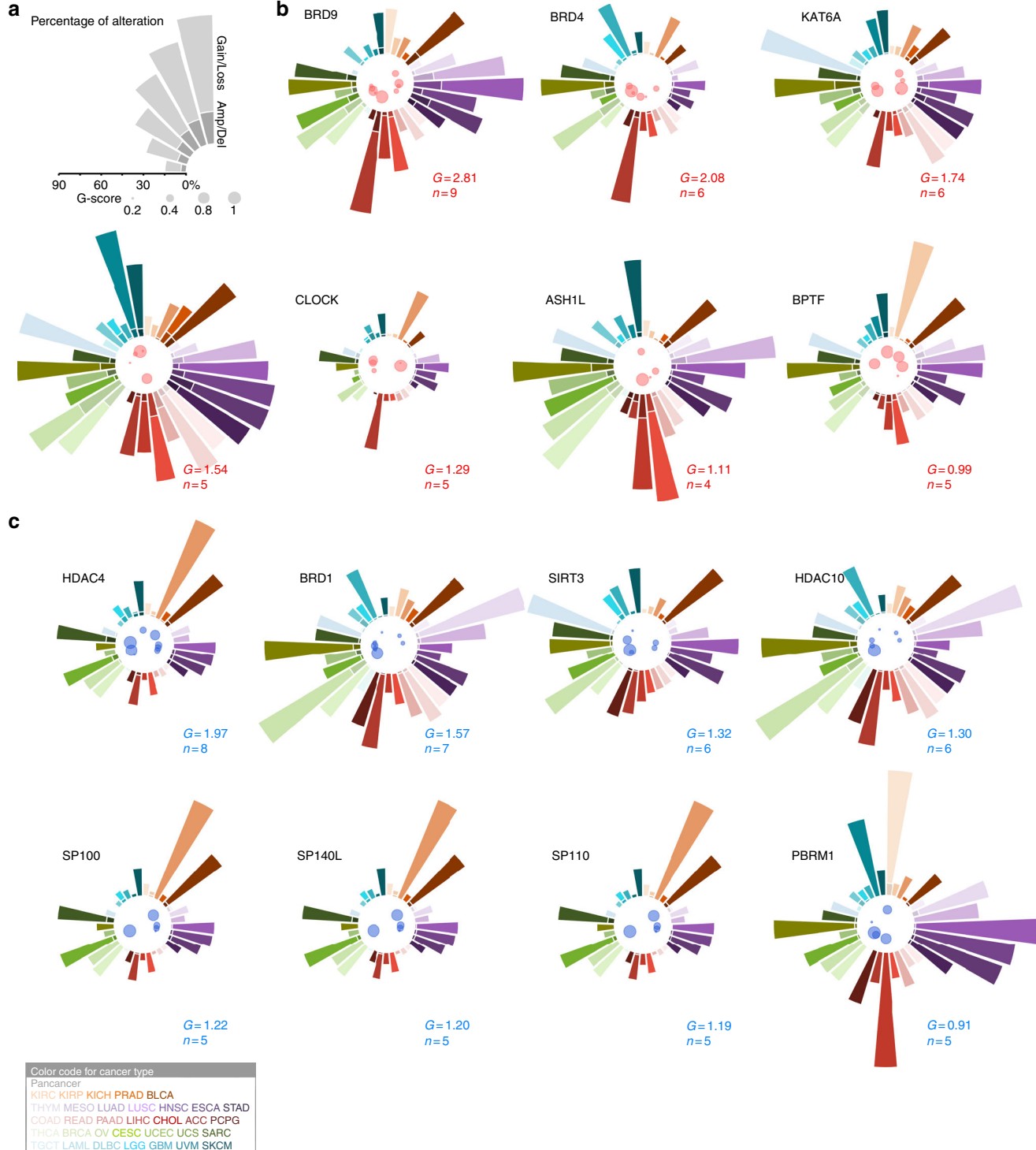

**Fig. 3** The putative cancer-causing HAMPs driven by SCNAs across cancers. **a** The scales of the alteration percentage (upper) and G-score (lower). **b, c** Seven and eight putative cancer-causing HAMPs that are commonly and recurrently gained (**b**) or lost (**c**), respectively, across 33 cancers (overall G-score > 0.9). In the center of each circle, the cancer types that harbored recurrent alterations of a given HAMP are indicated by a color-coded bubble (red: gain; blue: loss). The size of the bubble plot represents the G-score of the HAMP in a given cancer type. The bar diagram of each circle shows the percentage of gain/amplification (**b**) or loss/deletion (**c**) of certain HAMPs in individual cancer types. The cancer type is coded by color. Gain: GISTIC status = 1; amplification: GISTIC status = 2; loss: GISTIC status = − 1; and deletion: GISTIC status = − 2. The overall G-score and the number of cancer types that harbored recurrent alterations are indicated as G and n, respectively

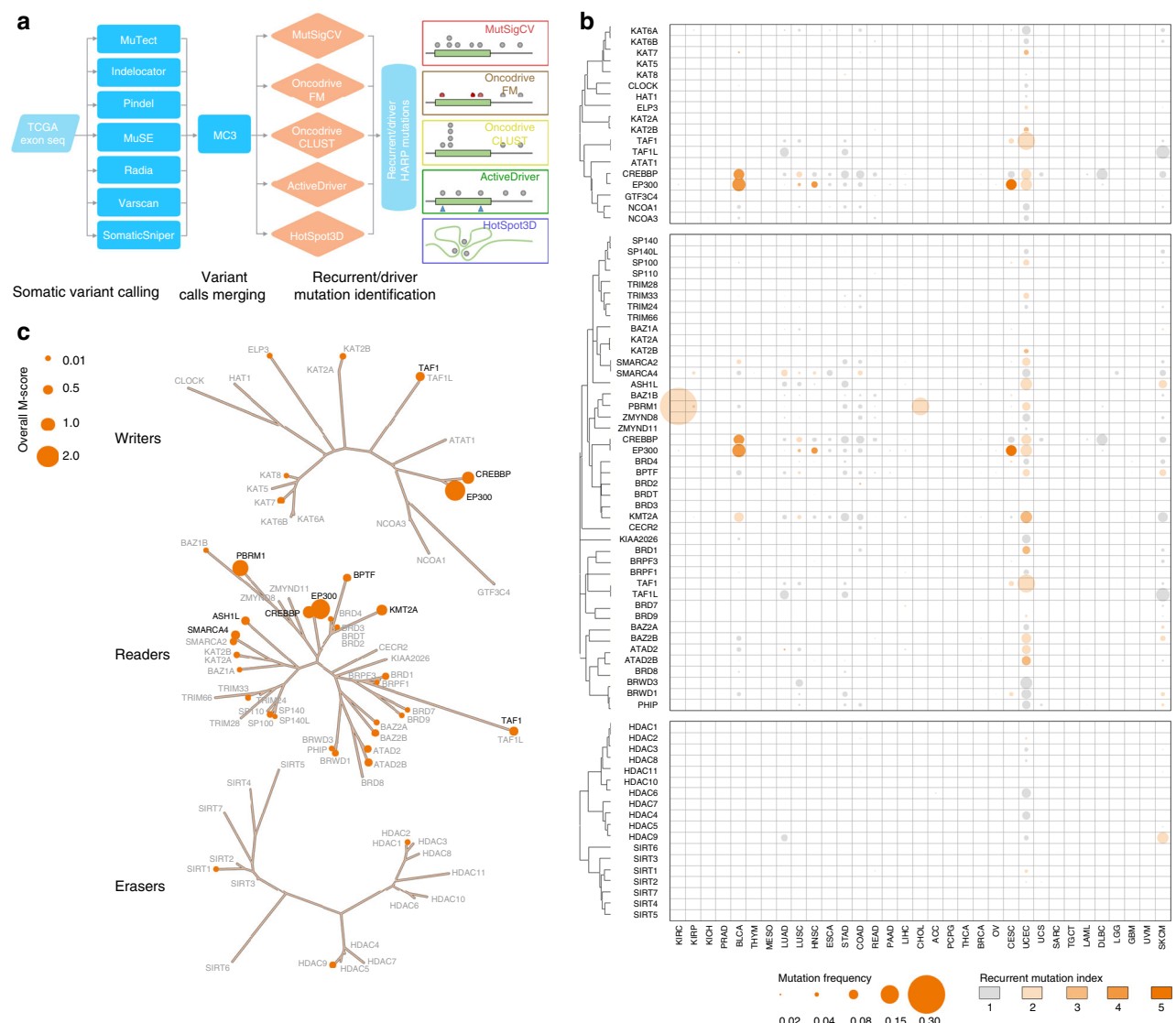

**Fig. 4** Somatic mutations of HAMPs across cancers. **a** The workflow of somatic mutation analysis. Seven algorithms were used for variant calling and five approaches were used for recurrent mutation identification. **b** The bubble plot shows the mutation frequencies and mutation indexes of the putative cancer-causing HAMPs driven by somatic mutations in each cancer type. The size of the bubble: mutation frequency; intensity of color: mutation index. The phylogenetic trees were generated by multiple sequence alignments of the full-length sequences of the proteins. **c** Summary of the overall M-scores of the putative cancer-causing HAMPs driven by somatic mutations, which were identified in at least one cancer type. The size of the bubble: the overall M-score. The HAMPs coding in black and gray indicate a gene with an overall M-score ≥ 0.4 and < 0.4, respectively. The phylogenetic trees were generated by multiple sequence alignments of the full-length sequences of the proteins

fusions of the same HAMPs with multiple different partners, *ASH1L*, *SMARCA4*, *BPTF*, *CREBBP*, and *KMT2A* were the most prominent and recurrent HAMPs, associated with 24, 20, 20, 18, and 17 fusion events, respectively (Fig. 6a and Supplementary Data 18). Among the three types of HAMPs, BRD proteins showed the highest frequency of transcript fusions, whereas HDACs had the fewest events (Fig. 6b). Notably, among the 9799 tumor specimens analyzed in our study, fusion events for HAMPs were only detected in 365/9799 (3.63%) tumors (Fig. 6c, d). Taken together, these results suggest that, although certain HAMP transcript fusions may be actionable for clinical cancer care, transcript fusion is a rare genomic alteration compared with SCNAs and mutations of HAMPs in common cancers.

**Putative therapeutic target HAMPs across cancers.** To estimate the recurrent alterations of HAMPs at a pan-cancer level, we

calculated an overall recurrent score via an unweighted numeric sum of the numbers of recurrent events for a given HAMP across all cancer types (Fig. 7a). We found that 63 HAMPs were recurrently altered at the genomic level in at least one cancer type (Fig. 7b and Supplementary Data 20). Among those HAMPs, *BRD9*, *PBRM1*, *EP300*, *HDAC4*, *ATAD2*, *BPTF*, *BRD1*, *BRD4*, and *SP100* had the highest recurrent scores. Across the different lineages of cancers, three types of alterations were observed: (1) consistent, putative gain-of-function (such as for *KAT6A* and *CLOCK*, which were focally amplified in six and five cancer types, respectively); (2) consistent, putative loss-of-function (such as for *PBRM1*, which is mutated in four and focally deleted in five cancer types, and for *CREBBP*, which is mutated in four and focally deleted in two cancer types); and (3) diverse alterations (such as for *ATAD2*, which is mutated in three and focally amplified in five cancer types, and for *BPTF*, which is mutated in three and focally amplified in five cancer

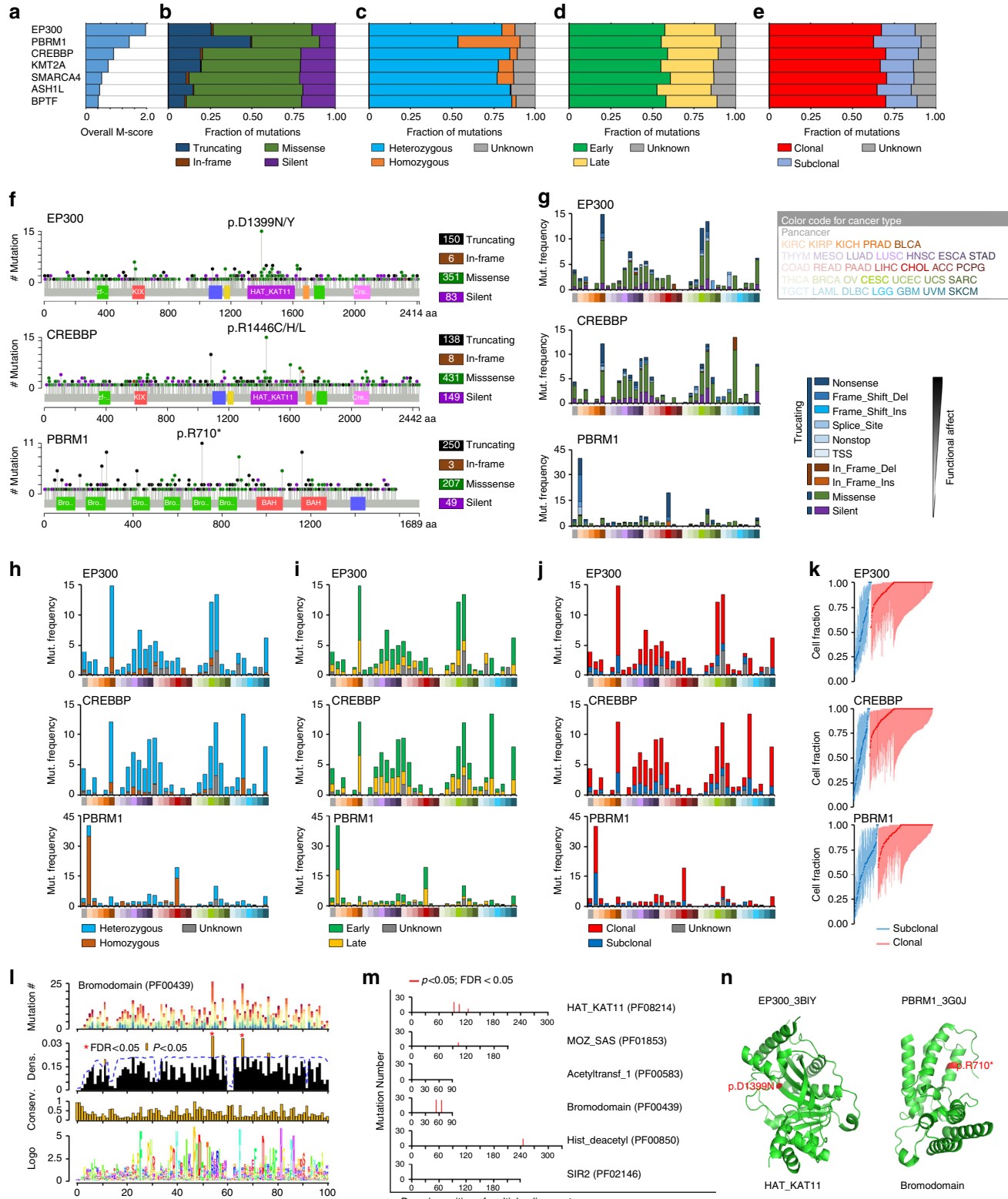

types). This suggests that a large fraction of HAMP alterations may be commonly shared by multiple cancer types, whereas others may be tumor-lineage dependent.

Next, we calculated the numbers of recurrently altered HAMPs for each individual cancer type and found that they are remarkably diverse among different tumors. UCEC, BLCA, LUSC, and SKCM had higher numbers of alterations, whereas KIRP and LAML had only one recurrent alteration, and DLBC,

KICH, PAAD, PCPG, THCA, and THYM did not have any therapeutic targets (Fig. 7c, Supplementary Figure 5, and Supplementary Data 21). Among the recurrent genomic events, those in *BRD9*, *EP300*, *ATAD2*, *HDAC4*, *PRBM1*, *BRD4*, and *BRD1* were observed in the highest numbers of cancer types (nine, eight, eight, eight, seven, seven, and seven, respectively) (Supplementary Figure 6 and Supplementary Data 20).

**Fig. 5** The putative cancer-causing HAMPs driven by somatic mutations across cancers. **a** Overall M-scores of the seven putative cancer-causing HAMPs (M-scores > 0.4) across all cancer types. **b**–**e** The fractions of the mutation categories (**b**), types (**c**), timing status (**d**), and clonal heterogeneity (**e**) of the seven HAMPs across all cancer types. **f** The lollipop plots illustrate the distribution and categories of somatic mutations in the *EP300*, *CREBBP*, and *PBRM1* gene-coding sequences across all cancer types. Note that although mutations are randomly distributed along the entire coding sequence, in both the *EP300* and *CREBBP* genes the most frequent mutations are located within the HAT catalytic domain. **g**–**j** The fractions of the mutation categories (**g**), types (**h**), timing status (**i**), and clonal heterogeneity (**j**) of *EP300*, *CREBBP*, and *PBRM1* in individual cancer types. **k** Clonal heterogeneity of *EP300*, *CREBBP*, and *PBRM1* across cancers. On the basis of the probability distributions of the cancer cell fractions, mutations were determined to be either clonal (red blocks) or subclonal (blue blocks). **l** An example of a meta-domain represented 51 different BRDs within BRD-containing genes. The *x*-axis shows the amino acid positions of the alignment. The *y*-axis, from top to bottom panel, indicates the mutation number of each position, the significance of a mutation in a given position (LowMACA), the amino acid conservation of the meta-domain, and the Pfam hidden Markov model sequence logo generated via the Skylign tool where the height of each stack of residues indicates the relative entropy for that position. **m** Bar plots showing the stacking of mutations within the six main HAMP Pfam domains. The *x*-axis depicts the position in the global alignment and the *y*-axis shows the mutation number of each position. The red bars highlight the residues that are significant according to the Benjamini–Hochberg procedure for multiple testing correction of *p*-values, which we performed on the conserved positions using LowMACA. **n** Ribbon drawings of the crystal structures of the HAT_KAT11 domain within EP300 (3BIY) and Bromodomain within PBRM1 (3G0J). The D1399 and R710 residues that were identified as hotspot mutations are shown in red

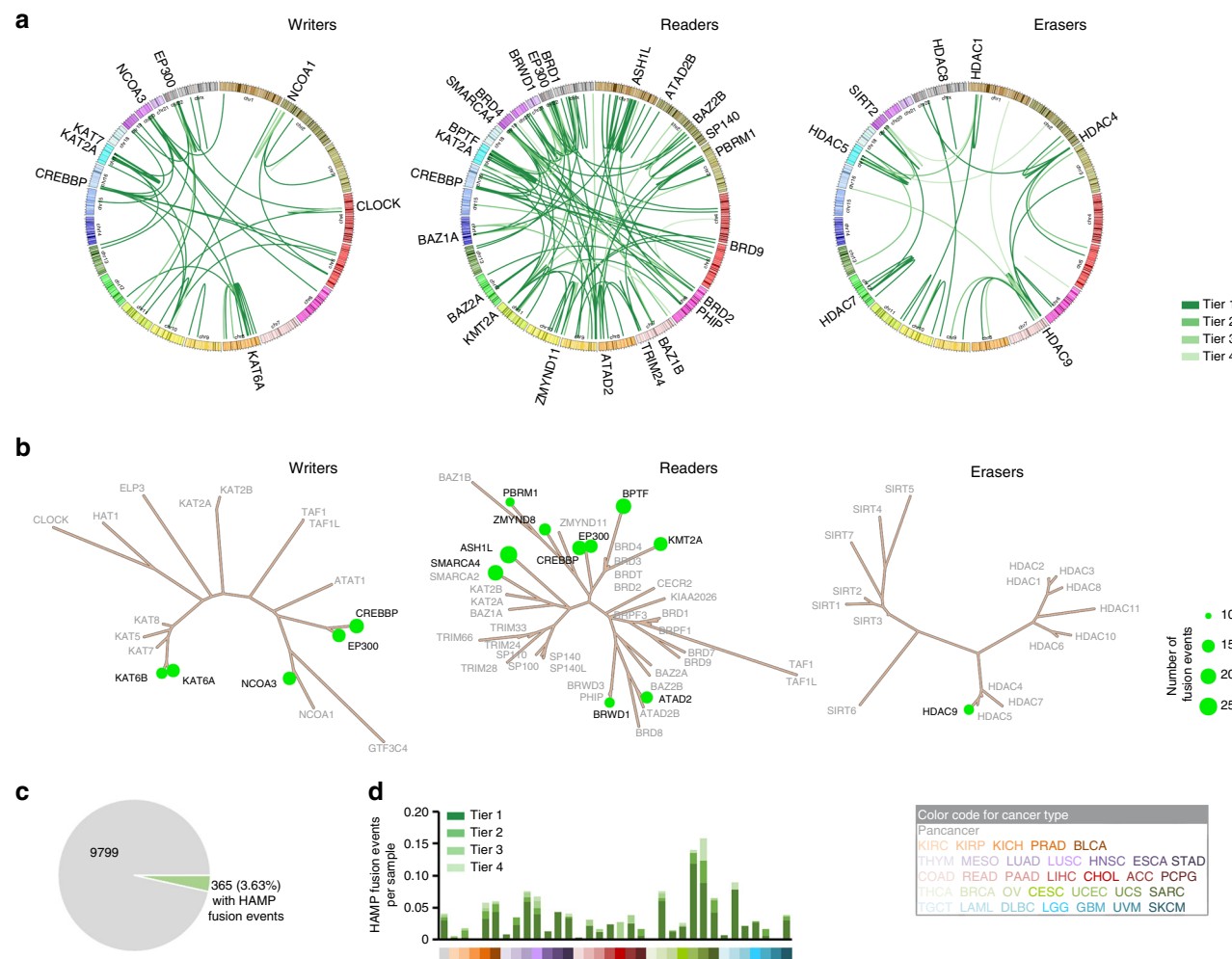

**Fig. 6** Transcript fusions of HAMPs across cancers. **a** The fusions of HAMPs in TCGA samples are shown in Circos plots. The fused genes are illustrated as lines that connect two parental genes. The intensity of the line corresponds to the tier of the fusion event. **b** Summary of the HAMP transcript fusions across cancers. The size of the bubble: number of the HAMP transcript fusion events across 33 cancer types. The HAMPs coding in black and gray indicate a gene with fusion events ≥ 10 and < 10, respectively. The phylogenetic trees were generated by multiple sequence alignments of the full-length sequences of the proteins. **c** Percentage of tumor specimens with HAMP transcription fusion events across all cancer types. **d** Percentage of tumor specimens with HAMP transcription fusion events in each cancer type

Finally, to survey whether the HAMPs with high recurrent scores had been characterized in physiological and pathological conditions, we performed a database search for related publications through PubTator using gene/protein names. We found that nearly two-thirds of the HAMPs had not been well characterized (i.e., PubTator score < 150), including many HAMPs with high recurrent scores (Fig. 7d). When we searched for HAMPs in the databases of patents (World Intellectual Property Organization)

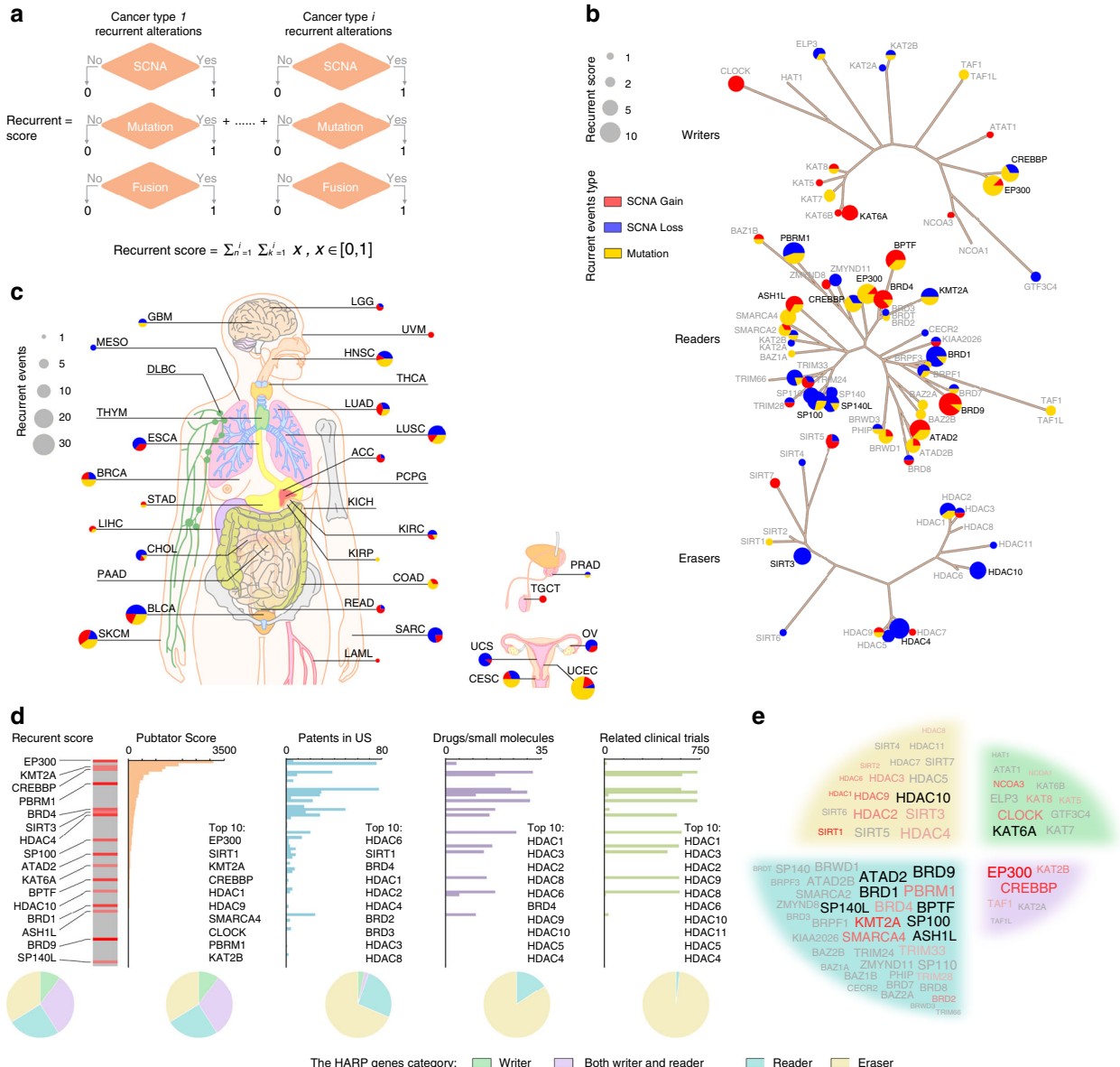

**Fig. 7** Putative therapeutic target HAMPs across cancers. **a** The workflow for the putative therapeutic target HAMP analysis. An unweighted numeric sum of the numbers of recurrent events for a given HAMP was calculated across all cancer types (referred to as the pan-cancer recurrent score). **b** Summary of the recurrent score of the putative cancer-causing HAMPs driven by recurrent genomic alterations. The size of the bubble: the overall recurrent score. Red: amplification; blue: deletion; yellow: mutation. The HAMPs coding in black and gray indicate a gene with overall recurrent score ≥ 6 and < 6, respectively. The phylogenetic trees were generated by multiple sequence alignments of the full-length sequences of the proteins. **c** Summary of the recurrent events of the HAMPs in each cancer type. The size of the bubble: total recurrent events. Red: amplification; blue: deletion; yellow: mutation. The phylogenetic trees were generated by multiple sequence alignments of the full-length sequences of the proteins. **d** Summary of the results of a database search for publications, patent applications, small molecule drugs, and clinical trials related to each HAMP. The HAMPs are ranked according to their PubTator scores. HAMPs with overall recurrent scores > 5 are highlighted as red bars. The color intensity represents the recurrent score. The pie diagrams at the bottom summarize the percentages of the HAMPs that are writer, both writer and reader, reader, and eraser genes for potential targets, publications, patent applications, small molecular drugs, and clinical trials. **e** A word cloud of the putative therapeutic target HAMPs identified in this study. The size of the font represents the recurrent score. The color of the font indicates the type of genomic alterations. Black: understudied targets (PubTator score < 150 and recurrent score > 5); red: well-studied targets (PubTator score ≥ 150 and recurrent score > 5; the intensity of red indicates the PubTator score); gray: potential targets (recurrent score ≤ 5)

and clinical trials (ClinicalTrials.gov), we found that patent applications, drug development, and clinical trials were narrowly focused on HAMPs that have been well characterized in academic laboratories, such as HDACs and BRD2/3/4 (Fig. 7d). We identified 57 drugs that are currently being investigated in 787 clinical trials with 16 HAMPs listed as primary targets (Supplementary Data 22-23). Among them, HDACs (11/16,

68.8%) are the most common targets for drug development and clinical trials. In contrast, a large percentage of the putative therapeutic target HAMPs are still not extensively characterized and lack chemical probes for targeting. For example, 66.7% (42/63) of putative therapeutic target HAMPs are largely uncharacterized with limited indications about how these genes influence cell biology and cancer, although many of these

understudied HAMPs are recurrently altered among cancers, such as *BRD9*, *BRD1*, *BPTF*, and *ATAD2* (Fig. 7e). Notably, although a large percentage of reader genes among HAMPs showed high recurrent scores, the research efforts into biological function and drug development for these genes are still relatively limited compared with those for eraser genes (Fig. 7d). Taken together, our findings indicate that functional studies are urgently needed for these putative therapeutic target HAMPs, which may stimulate novel epigenetic therapeutic approaches.

**Identification of *BRD9* as a potential therapeutic target**. Our integrated genomic analysis identified 63 potential therapeutic target HAMPs, many of which (42/63) are understudied proteins according to their PubTator scores (<150). We hypothesized that well-annotated genomic and clinical information from TCGA may serve as a powerful resource to guide functional characterization of those candidate genes. We chose *BRD9* as a candidate to prove this concept, because it is an understudied HAMP gene with the highest recurrent score in our analysis (Supplementary Figure 7). To perform a clinically relevant analysis, we focused on nine cancer types in which *BRD9* is focally amplified. First, we analyzed *BRD9* mRNA expression and found that *BRD9* mRNA expression levels were significantly higher in the *BRD9*-amplified tumors than in the non-amplified tumors across all nine cancer types (Fig. 8a). We also compared *BRD9* mRNA expression between tumors and their corresponding control specimens from TCGA, except ACC, OV, and CESC, for which the RNA-seq profile from a normal control is unavailable (ACC and OV) or insufficient for sample numbers (CESC, $n = 3$) at TCGA. We found that *BRD9* mRNA was significantly upregulated in all cancer types studied (Fig. 8a). Identical results were also observed at a pan-cancer level (Fig. 8a). We confirmed *BRD9* protein expression in cancers by retrieving the immunohistochemical staining data from the Human Protein Atlas[26]. Nuclear staining of *BRD9* protein was detected in more than 90% of tumor specimens (Fig. 8b), which is consistent with its predicted nuclear function.

We proposed that the functions of a poorly characterized epigenetic regulator may be predicted on the basis of the known functions of genes that are co-expressed. Thus, Guilt-by-association (GBA) analysis may be particularly useful for gleaning an understanding of *BRD9* functions, given that *BRD9* is a subunit of the SWI/SNF complex[27–29]. As TCGA provides multi-omic profiles of large-scale sample sets, it is an excellent resource for GBA-based function prediction. We ranked the genes whose expression were positively correlated with *BRD9* at both the cancer type-specific and pan-cancer levels (Fig. 8c), then performed gene-set enrichment analysis. Among the 44 pathways identified, eight pathways were commonly shared by most cancer types ($n > 5$) and 21 pathways appeared to be enriched in certain cancer types (Fig. 8d and Supplementary Data 24). The biological processes that were most over-represented for *BRD9*-associated genes across nine cancer types were the cell cycle, DNA damage repair, and RNA metabolism pathways (Fig. 8e and Supplementary Data 24). Interestingly, the SWItch/Sucrose Non-Fermentable (SWI/SNF) complex has been reported to regulate genes involved in cell cycle and DNA damage repair[30–33]. To further test the function of *BRD9* in cancer, we knocked down *BRD9* expression in four breast and ovarian cancer cell lines with short hairpin RNAs (shRNAs) (Fig. 8f), then analyzed cell growth in both anchorage-dependent (Fig. 8g) and -independent (Fig. 8h) conditions. Consistent with the GBA prediction, knocking down *BRD9* expression dramatically inhibited cancer cell growth in vitro (Fig. 8f, g). Finally, we demonstrated that the expression of *BRD9* shRNAs significantly suppressed the growth of subcutaneous tumors formed by MDA-MB-231 or OVCAR8 cells in nude mice (Fig. 8j–m). Collectively, our results demonstrate that genetic depletion of *BDR9* expression significantly represses tumor cell growth in vitro and in vivo, suggesting that small molecular compounds targeting *BRD9* may have strong clinical application potentials.

## Discussion

Given that the genes that modulate the epigenome are altered in cancer at unexpectedly high frequencies, there is great interest in the development of therapeutic approaches that effectively target cancer epigenomes[5–8,12,34]. Thus, a systematic analysis of the epigenome based on the integration of multi-dimensional genomic profiles in large patient cohorts is urgently needed. TCGA provides a powerful resource to analyze the recurrent genomic alterations of the epigenome-modifying genes across different tumor lineages and develop an integrated view of the commonalities and differences. In the present study, by integrating various computational algorithms, we systematically characterized the genomic alterations of HAMPs across TCGA patients ($n = 11,193$, including 33 cancer types from 27 primary sites) at both the pan-cancer and cancer type-specific levels. We developed a publicly accessible database (FCG data portal: http://52.25.87.215/home/) to assist researchers with analyzing and visualizing the expression and genomic alterations of HAMPs in cancer. In our study, an overall recurrent score for each HAMP was estimated at a pan-cancer level by an unweighted numeric sum of the numbers of recurrent events. This overall recurrent score can prioritize the potential candidates for future anticancer drug development. A HAMP with a high score indicates its recurrent alterations are common genomic events across multiple adult cancer types. Thus, successful strategies targeting such HAMPs may benefit relatively larger proportions of patients from different cancer types. For example, *BRD9* shows highest overall recurrent score among HAMPs (e.g., focally amplified in nine cancer types), strongly suggesting that targeting *BRD9* by small molecular inhibitors may be a novel treatment strategy with wide clinical application for cancer therapy. Finally, given that many HAMPs with high scores are understudied genes (e.g., *BRD9*), the overall recurrent score may also be used to prioritize the genes for basic biological studies in context of cancer.

Our expression analysis showed that most HAMPs were ubiquitously expressed across all cancer types. However, the HAMP expression signatures differentiated the specimens from different cancer types and grouped together the tumors with related lineage origins. This finding suggests that cancer type-dependent acetylation status should be considered when designing treatments to target histone acetylation in cancer. Interestingly, we also identified eight HAMPs with expression restricted to certain cancer types, which may be exploited for potential cancer-specific HAMP-targeted approaches. For example, we observed that *BRDT* was specifically expressed in testicular cancer and a small proportion of lung, esophageal, and uterine cancers. Next, consistent with a previous pan-cancer analysis[19], our SCNA analysis showed that a large percentage of HAMPs had recurrently altered DNA copy numbers. For example, the BRD proteins *BRD9*, *BRD4*, and *ATAD2* were focally amplified in multiple cancer types and their increased copy numbers were significantly, positively correlated with higher mRNA expression levels. Given that BRD proteins are putative druggable proteins[6,7,11,35–37], our results provide clinically relevant targets for future anticancer drug discovery. Notably, several HDAC genes, such as *HDAC4* and *HDAC10*, showed significant, focal copy number loss, which is consistent with the experimental evidence that certain HDAC genes may serve as tumor suppressors during tumorigenesis.

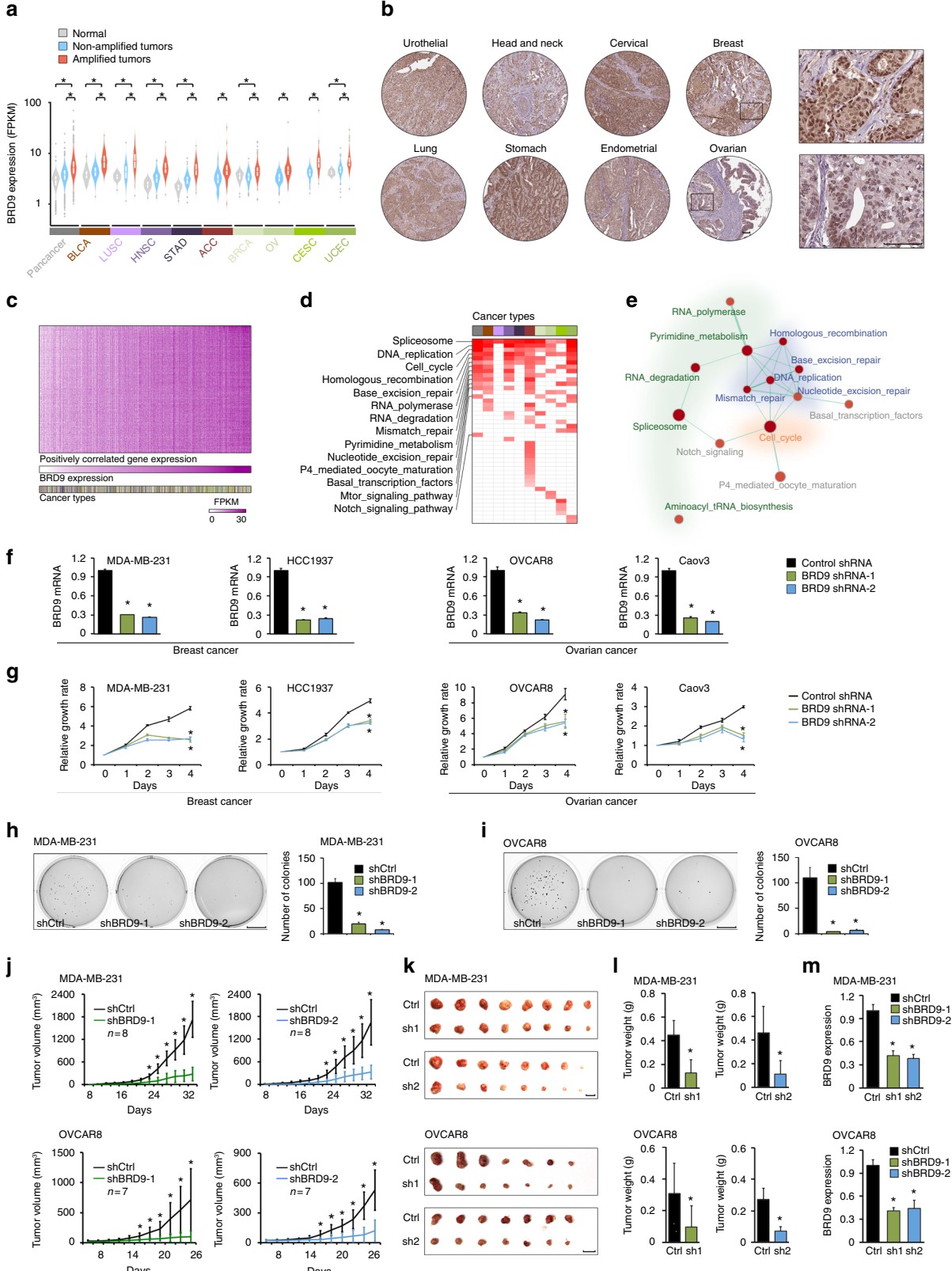

Furthermore, genetic depletion of HDACs in tumor cells leads to cell cycle arrest, apoptosis, and senescence, suggesting that HDACs are required for the survival and growth of tumor cells[4,9,10]. Collectively, our findings indicate that targeting select HDAC isoforms in certain cancer types may be a more precise approach for the treatment of cancer compared to the use of pan-HDACis. Third, our mutation analysis showed that HAMPs were recurrently mutated in cancers with different behaviors. For example, *CREBBP* and *EP300* were widely mutated across multiple cancer types at relatively low frequencies; most of these mutations were heterozygous missense mutations. In contrast, *PBRM1* showed high-frequency, cancer type-specific mutations in

**Fig. 8** Identification of *BRD9* as a potential therapeutic target for cancer treatment. **a** Expression levels of *BRD9* mRNA in nine cancer types in which *BRD9* was focally amplified. Gray: normal controls; white: tumors without *BRD9* amplification; red: tumors with *BRD9* amplification. *BRD9* copy number status in each tumor specimen was estimated by a GISTIC analysis. **b** Expression of *BRD9* protein was retrieved from the Human Protein Atlas dataset. A typical immunohistochemical staining of each cancer type is presented. Scale bar indicates 100 μm. **c** The heatmap of the genes that were significantly, positively co-expressed with *BRD9* in nine cancer types. Each tumor is represented in a column and each gene in a row. The RNA expression of *BRD9* is presented as a bar graph under the heatmap. The genes were arranged from top to bottom in descending order of their correlation with *BRD9*. **d** The pathways over-represented for *BRD9*-associated genes at the pan-cancer and individual cancer levels, according to pre-ranked gene-set enrichment analysis. **e** The functional hubs of pathways associated with *BRD9* expression as analyzed by pre-ranked gene set enrichment. **f** *BRD9* was knocked down with two independent shRNAs in four different breast and ovarian cancer cell lines. Knockdown efficiency was confirmed by real-time RT-PCR. **g** Four-day growth curves of cancer cell lines transfected with control or *BRD9*-specific shRNAs. **h, i** Soft agar assays of MDA-MB-231 (**h**) and OVCAR8 (**i**) cells expressing control and *BRD9*-specific hairpins. Scale bar indicates 1 cm. Quantification of the number of colonies from the soft agar assays on MDA-MB-231 (**h**) and OVCAR8 (**i**) cells. **j** In vivo xenograft tumor growth curve of MDA-MB-231 (upper) and OVCAR8 (lower) cells expressing control and *BRD9*-specific hairpins. **k** Representative pictures of tumors derived from control and *BRD9*-specific hairpin-expressing cells. Scale bar indicates 1 cm. **l** In vivo tumor weight curves for MDA-MB-231 (upper) and OVCAR8 (lower) cells expressing control and *BRD9*-specific hairpins. **m** Knockdown efficiency of xenograft tumors was confirmed by real-time RT-PCR. *t-test p-value < 0.05

KIRC (40.1%). Importantly, a high percentage of homozygous truncating mutations were observed in the *PBRM1* gene in KIRC. Finally, it has been reported that transcript fusions of HAMPs occur frequently in certain cancer types, which provides a rationale for targeted cancer therapy. For example, *BRD4* or *BRD3* can be fused to the coding sequence of *NUTM1*, to create a chimeric gene that encodes the BRD–NUT fusion protein in about 75% of NUT midline carcinomas, a rare squamous cell epithelial cancer[38]. However, our transcript fusion analysis indicates that fusion events are rare genomic alterations to HAMPs in common adult cancers. Among the 9799 specimens analyzed, we observed only 400 HAMP fusion events, including 298 tier 1 fusion events, which suggests that targeting HAMP fusion proteins in common adult cancers may have limited clinical application.

Collectively, our comprehensive genomic analysis identified 63 putative cancer-causing HAMPs driven by SCNAs and/or mutations (recurrent score ≥ 1). Among the 15 HAMPs with copy numbers that were significantly altered in cancer (overall G-score > 0.9), 7 had recurrently, focally increased copy numbers, which suggests that they may function as oncogenes during tumorigenesis. Thus, the development of potent and specific inhibitors to directly target these amplified HAMPs may be a therapeutic approach to treat certain cancer types identified in our study. For example, the breast and ovarian cancer patients with *BRD4* amplifications may be potential candidates for treatment with BET inhibitors that have been successfully developed to the preclinical stage[39,40]. In addition, eight HAMPs focally lost copy numbers (overall G-score > 0.9) and eight HAMPs showed high frequencies of mutations (overall M-score > 0.4), which indicates that their functions may be reduced and/or deficient due to partial loss of wild-type alleles during cancer development. These cancer-specific vulnerabilities may represent tractable therapeutic opportunities via the induction of synthetic lethality. For example, mutations of the BRD-containing protein *SMARCA4* in tumor cells results in a unique functional dependence on *SMARCA2*. Thus, *SMARCA2* serves as a potential therapeutic target for *SMARC4*-mutant cancers[41]. Notably, very few homozygous deletions or mutations of HAMPs (except for *PBRM1*) were observed in cancer, suggesting that HAMPs may be essential for tumor growth, and that complete loss may be lethal. Therefore, beyond the synthetic lethal approach, hemizygous loss of HAMP genes may render tumor cells highly dependent on the remaining wild-type allele. This vulnerability in tumor cells may present potential therapeutic opportunities. For example, tumor cells may be more sensitive than normal cells, which have two copies of the genes, to inhibitors that target these HAMPs.

Among the three groups of HAMPs, HDACs are the best-characterized gene family and their inhibitors have been approved to treat hematopoietic malignancies[5–10,34]. Although HDACis have also been widely tested in solid tumors in early clinical phases, the results from the clinical trials have revealed limited anticancer activity. Our results indicate that certain HDAC genes (e.g., *HDAC4* and *HDAC10*) focally lose copy numbers in selected cancer types. The copy numbers lost in tumors may depend on the remaining HDAC allele; thus, further suppression of HDACs may lead to greater anticancer effects in these tumors. In addition, given that different HDAC genes have distinct genomic alterations in cancer, the development of isoform-selective HDACis is urgently needed for future clinical application. Importantly, a large portion of putative cancer-causing HAMPs identified in our study are understudied HAMPs (PubTator Score < 150) that have not been targeted with chemical compounds or Food and Drug Administration-approved drugs. This study provides promising candidate genes for both basic cancer research and medical chemistry development. For example, well-ordered, deep, hydrophobic pockets in BRDs provide a highly favorable locus for the binding of small molecule compounds. We identified 15 BRD genes that may function as oncogenes in cancers, including *BRD9*, *BRD4*, and *ATAD2*. Excitingly, a number of inhibitors that target *BRD4* have been developed[42,43] and evaluated in multiple preclinical cancer models[40,43–54].

Among 63 potential therapeutic target HAMPs identified in our study, *BRD9* is one of the most promising understudied candidates. Our genomic and functional studies demonstrated that: (1) *BRD9* is recurrently and focally amplified in nine cancer types with the highest recurrent score; (2) the expression of *BRD9* is significantly increased in cancer specimens compared with that in corresponding normal tissues; (3) computational prediction suggests that *BRD9* expression is associated with cell cycle, DNA damage repair, and RNA metabolic pathways in cancer; and (4) genetic depletion of *BRD9* by shRNAs reduced cancer cell growth in vitro and in vivo. Furthermore, although *BRD9* is an understudied epigenetic regulator with limited functional characterization (< 35 publications in PubMed), evidence from recent independent studies also strongly suggests its potential roles in cancer[28,29]. *BRD9* is a subunit of the SWI/SNF complex[27–29,33], which is highly altered in cancer genomes and has both tumor suppressor and oncogenic roles[30–33]. Certain types of hematologic cancer cells (e.g., acute myeloid leukemia cells) require *BRD9* to sustain *MYC* transcription, cell proliferation, and a block in differentiation[28]. Most importantly, due to its BRD, with its well-ordered, deep, hydrophobic pockets, *BRD9* is a druggable protein. Several small molecule compounds have been recently developed to target *BRD9*; they have shown promising anticancer effects in preclinical models[28,55–63]. Taken together, our functional studies

on *BRD9* are the proof-of-concept for the HAMP candidates identified in our study and strongly suggest the clinical potential of targeting *BRD9* in common adult solid cancers.

## Methods

**The HAMP family gene definition**. The HAMP gene family members ($n = 73$) were defined based on a review article[1] and were further complemented by database searching via the Human Protein Reference Database (http://www.hprd.org/), the Pfam protein family database (http://pfam.xfam.org/), and the SMART (Simple Modular Architecture Research Tool) database (http://smart.embl-heidelberg.de/).

**RNA-seq data processing and gene expression analysis**. The poly(A)$^+$ RNA-seq (Illumine) data were generated by the University of North Carolina and the British Columbia Cancer Agency Genome Sciences Centre as part of the TCGA project, and were processed by the TCGA Research Network using the NCI Genomic Data Commons (GDCs) mRNA quantification analysis pipeline (https://docs.gdc.cancer.gov/Data/Bioinformatics_Pipelines/Expression_mRNA_Pipeline/). The gene-level RNA expression data (in fragments per kilobase of transcript per million mapped reads [FPKM]) of tumor specimens across 33 cancer types from 27 primary sites, as well as corresponding normal adjacent specimens from 24 matched tissue types were downloaded from the GDC Data Portal (https://portal.gdc.cancer.gov/) (retrieved date: 27 October 2017). If more than one sample existed for a participant, one single tumor sample (and matched adjacent sample, if applicable) was selected based on the following rules: (1) tumor sample type: primary (01) > recurrent (02) > metastatic (06); (2) order of sample portions: higher portion numbers were selected; and (3) order of plate: higher plate numbers were selected. The gene expression data of 2012 cancer cell lines were downloaded through the Expression Atlas (https://www.ebi.ac.uk/gxa/download.html) under the accessions E-MTAB-2770, E-MTAB-3983, and E-MTAB-2706.

**SNP array data collection and processing**. The SNP array data (Affymetrix Genome-Wide Human SNP Array 6.0) in CEL format across 33 cancer types were downloaded from the TCGA Data Portal (https://tcga-data.nci.nih.gov/tcga/). Segmentation files of TCGA tumor samples processed by circular binary segmentation algorithm[64] were retrieved from the TCGA GDAC Firehose of the Broad Institute (http://gdac.broadinstitute.org/; retrieved date: 3 January 2018). If multiple samples existed for one participant, one pair of tumor and matched control was selected for ABSOLUTE analysis and one tumor sample was kept for focal SCNA analysis. Sample selection based on following rules: (1) sample type: for tumor tissues, primary (01) > recurrent (02) > metastatic (06); for normal control tissues, blood (10) > solid (11); (2) molecular type of analyte for analysis: prefer D analytes (native DNA) over G, W, or X (whole-genome amplified); (3) order of sample portions: higher portion numbers were selected; and (4) order of plate: higher plate numbers were selected.

**Recurrent focal SCNA estimation**. The Genomic Identification of Significant Targets in Cancer (GISTIC 2.0) algorithm[20] (https://www.broadinstitute.org/cancer/cga/gistic) was used to identify significantly recurrent focal genomic regions that were gained or lost in a given tumor type. Segmentation files retrieved from the TCGA GDAC Firehose of the Broad Institute were used as input. GISTIC deconstructed copy number alterations into broad and focal events and applied a probabilistic framework to identify location and significance levels of SCNA. For the recurrent focal SCNA estimation, the significance levels ($q$-values) were calculated by comparing the observed gains/losses at each locus to those obtained by randomly permuting the events along the genome. Tumors which had more than 2000 segments were excluded from our analysis. Default parameters of GISTIC were used with the confidence level set to 0.99 (by -conf). Focal events with $q$-value below 0.25 were considered as significantly recurrent. Significant focal events in individual samples were then classified into four categories according to the amplitude threshold of GISTIC: GISTIC status = 0, below threshold; GISTIC status = 1, amplified (gain); GISTIC status = 2, highly amplified (amplification); GISTIC status = −1, deleted (loss); GISTIC status = −2, highly deleted (deletion). In each cancer type, a GISTIC score (G-score), which counts in both frequency and amplitude of a given SCNA event[20], was generated for each HAMP gene for gain or loss separately. The gene with a G-score < 0.1 was excluded from downstream analysis due to its low frequency and/or amplitude. For a given HAMP gene, an overall G-score across all cancer types was calculated by an unweighted sum of G-scores in every cancer type.

**Correlation analysis between copy number and RNA expression**. To identify HAMP genes that had positive correlations between their RNA expression levels and copy number alterations, the putative gene-level copy number of a given gene was estimated by the GISTIC algorithm. The HAMP genes that were detectable in at least 10% of tumor specimens (90th percentile of FPKM value ≥ 1) in a given cancer type were subjected to correlation analysis. Pearson's correlation analysis was performed by R software and the threshold of significant correlation between the estimated copy number and RNA expression level for each gene was set to $p < 0.001$ (Pearson's correlation).

**Identification of the putative cancer-causing HAMPs driven by SCNAs**. At the individual cancer type level, we identified the putative cancer-causing HARP genes driven by SCNAs using four criteria as follows: (1) located in a peak region of a significantly recurrent focal SNCA locus estimated by GISTIC ($q \leq 0.25$); (2) altered with high frequency and large amplitude (G-score ≥ 0.1); (3) mRNA expression was reliably detected in at least 10% of tumor specimens in a given cancer type (the 90th percentile of FPKM value ≥ 1); and (4) expression level of mRNA was significantly and positively correlated with the estimated copy numbers ($p$-value of Pearson's correlation coefficient between log[FKPM + 1] and logR < 0.001). To estimate SCNAs for these putative cancer-causing HARP genes at a pan-cancer level, we calculated an overall G-score by an unweighted numeric sum of G-scores that met all four criteria in each individual cancer type.

**Whole-exome sequencing data collection and processing**. Mutation Annotation Format (MAF) profiles for 33 cancer type were downloaded from the TCGA MC3 project (https://doi.org/10.7303/syn7214402), a variant calling project of TCGA[21]. The MC3 data were generated through seven independent mutation calling algorithms, including Pindel (INDEL), MuSE (SNV), Radia (SNV)[65], VarScan2 (SNV/INDEL), MuTect (SNV), Indelocator (INDEL), and SomaticSniper (SNV). Variants from each caller were merged, quality control filtered, and stored in MAF file[21]. If multiple samples existed for a participant in the MAF, one single pair of tumor/matched control sample was kept following the rules: (1) sample type: for tumor tissues, primary (01) > recurrent (02) > metastatic (06); for normal tissues, blood (10) > solid (11); (2) molecular type of analyte for analysis: prefer D analytes (native DNA) over G, W, or X (whole-genome amplified); (3) order of sample portions: higher portion numbers were selected; and (4) order of plate: higher plate numbers were selected. We excluded all mutations that were not tagged with PASS or WGA alone in all cancer types.

**Recurrent mutation gene estimation**. To predict the putative cancer-causing HAMP genes driven by mutation, five independent methods were integrated and applied to identify recurrent mutations: (1) MutSigCV (http://software.broadinstitute.org/cancer/software/genepattern/modules/docs/MutSigCV), which identifies genes that are significantly mutated in cancer genomes, using a model with mutational covariates. It analyzes the mutations of each gene to identify genes that were mutated more often than expected by chance, given the background model; (2) Oncodrivefm (http://bg.upf.edu/group/projects/oncodrive-fm.php), which computes a metric of functional impact using three well-known methods (SIFT, PolyPhen2, and MutationAssessor) and assesses how the functional impact of variants found in a gene across several tumor samples deviates from a null distribution to detect candidate driver genes; (3) OncodriveCLUST (http://bg.upf.edu/group/projects/oncodrive-clust.php), which is designed to exploit the feature that mutations in cancer genes, especially oncogenes, often cluster in particular positions of the protein and change their functions, thus used to nominate novel candidate driver genes; (4) ActiveDriver (http://reimandlab.org/software/activedriver/), which identifies posttranslational modification sites in proteins (i.e., active sites such as signaling sites, protein domains, regulatory motifs) that are significantly mutated in cancer genomes; and (5) HotSpot3D (https://github.com/ding-lab/hotspot3d), which identifies mutation hotspots from linear protein sequence and correlate the hotspots with known or potentially interacting domains and mutations. MC3 MAF files were used as input for the above programs and default parameters were used for all five programs. A mutation index $x$ (ranges from 0 to 5) was assigned to a gene, which has passed the threshold of $x$ out of five programs in a given cancer type. In addition, a mutation score (M-score) was calculated for each mutated HAMP gene in a given cancer type, which take into account both the mutation index and its frequency of mutation across samples (i.e., M-score = mutation index × mutation frequency). Genes with mutation index ≥ 2 (identified as positive by at least two programs) were considered as recurrently mutated. An overall M-score was generated to measure the recurrent mutation level of a given HAMP gene across all cancers, by unweighted sum of M-scores estimated for each individual cancer type.

**Intra-tumor genetic heterogeneity analysis**. The intra-tumor heterogeneity was estimated by ABSOLUTE algorithm[22] (http://archive.broadinstitute.org/cancer/cga/absolute). ABSOLUTE calculated the purity, ploidy, and absolute DNA copy numbers from the segmented copy number alterations and mutation profiles of tumor samples. HAPSEG package[66] was used to generate copy number data segmented by haplotype for each cancer type using TCGA SNP array data (Affymetrix Genome-Wide Human SNP Array 6.0) as input files. The default setting of HAPSEG was applied, except that the minimum segment size and outlier probability was set to 5 and 0.001, respectively. The output segmented copy ratios data of HAPSEG together with filtered MC3 MAF files were passed to ABSOLUTE for analysis. The parameters for ABSOLUTE were as follows: sigma.p = 0, max.sigma.h = 0.02, min.ploidy = 0.95, max.ploidy = 10, max.as.seg.count = 1500, max.non.clonal = 0, max.neg.genome = 0, platform = "SNP_6.0", and copy_num_type = "allelic". The cancer cell fraction of each somatic single-nucleotide variant was extracted from the output. Mutations were classified as clonal/subclonal and heterozygous/homozygous as reported by ABUSOLUTE. Mutation timing was inferred as previously reported[67]. Briefly, mutations were classified as

early or late based on their clonal status and mutation copy numbers. In general, clonal mutations represent relatively early events occurring before or at the time of the most recent clonal expansion, whereas subclonal mutations represent later events. In the context of genome-doubling or amplification, a mutation occurring before doubling would be expected to have multiple copies, whereas a mutation occurring after doubling will likely have only one copy. Thus, to timing mutations relative to copy number events, mutations with mutation copy number > 1 were classified as before event and any mutations with a mutation copy number of 1 were classified as after event. Integrating this with our clonal/subclonal status estimation, all clonal mutations that were classified as before event were assigned to early and all subclonal or after event mutations were assigned to late.

**Domain mutation analysis**. Domains that shared by HAMP genes were analyzed by the R package LowMACA[23] (https://www.bioconductor.org/packages/release/bioc/html/LowMACA.html). LowMACA was used to analyze the mutation profile of multiple proteins via consensus alignment and identify their mutational hotspots. The HAMP genes along with their associated Pfam domains (HAT_KAT11-PF08214, MOZ_SAS-PF01853, Acetyltransf_1-PF00583, Bromodomain-PF00439, Hist_deacetyl-PF00850, and SIR2-PF02146) and non-silent mutation information retrieved from the TCGA MC3 MAF file were used as the input. Default parameters were used for LowMACA analysis. ZMYND11 residues 122–203 within domain PF00439 and HDAC11 residues 23–320 within domain PF00850 were excluded from the analysis, because the maximum similarity with any other sequence was < 20%.

**Transcript fusion data collection and analysis**. The gene fusion data of TCGA were retrieved from TumorFusions data portal (http://tumorfusions.org/), which analyzed transcript fusions across 33 cancer types from TCGA[25]. The transcript fusion events were called by Pipeline for RNAseq Data Analysis[68] and fusions detected in normal samples were excluded. Six filters controlling for sequence similarity of the partner genes, transcriptional allelic fraction, dubious junctions, germline events, and the presence in non-neoplastic tissue were applied[25]. If more than one sample existed for a participant, one single sample was kept following the rules: (1) sample type: for tumor tissues, primary (01) > recurrent (02) > metastatic (06); (2) order of sample portions: higher portion numbers were selected; and (3) order of plate: higher plate numbers were selected. The genome-wide view (Circos plot) of transcript fusion events in HAMP genes was generated by Circos (http://circos.ca/).

**Recurrent gene score estimation**. The recurrent gene score was defined to estimate to what extent the HAMP gene could be cancer-driving at a pan-cancer level. The recurrent genomic alterations of each gene were integrated to assess its cancer-driving potential. In a given cancer type, each time when a HAMP gene was identified having a recurrent event (recurrent focal SCNA, recurrent mutation, or recurrent fusion), the gene was assigned a number 1, otherwise a number 0 was assigned. Then each HAMP gene will have a recurrent score ranged from 0 to 3 according to how many recurrent events were identified in a given cancer type. The overall recurrent gene score for a given gene was calculated by unweighted sum of its recurrent scores across all cancer types. The higher the score, the more likely this gene will be a putative cancer-causing genes.

**Cell culture**. Cancer cell lines were purchased from the ATCC or NCI Development Therapeutics Program. All cancer cell lines were cultured in RPMI1640 medium (Invitrogen) supplemented with 10% fetal bovine serum (VWR).

**shRNA knockdown and lentiviral transduction**. The pLKO.1 empty vector was purchased from Open Biosystms. Lentiviral shRNAs targeting BRD9 were constructed. Non-target shRNA (SHC002) was used as controls. Lentiviral vectors and packaging vectors were transfected into the packaging cell line 293T (ATCC) using the FuGENE6 Transfection Reagent (Promega). The media was changed 8 h post transfection and the media containing lentivirus was collected 48 h later. Cancer cells were infected with lentivirus in the presence of 8 μg/ml polybrene (Sigma). Knockdown efficiency was detected 72 h after infection by quantitative reverse-transcriptase PCR (qRT-PCR). The shRNA sequences are listed below:

BRD9 sh1:
F:5′-CCGGCCTGGATATTCAATGATAATACTCGAGTATTATCATTGAA
TATCCAGGTTTTTG-3′
R:5′-AATTCAAAAACCTGGATATTCAATGATAATACTCGAGTATTATC
ATTGAATATCCAGG -3′
BRD9 sh2:
F:5′-CCGGCAAGTCAGTTACGGAATTTAACTCGAGTTAAATTCCGTAA
CTGACTTGTTTTTG -3′
R:5′-
AATTCAAAAACAAGTCAGTTACGGAATTTAACTCGAGTTAAATTCCGTA
ACTGACTTG-3′
Control sh:
F:5′-CCGGCAACAAGATGAAGAGCACCAACTCGAGTTGGTGCTCTT
CATCTTGTTGTTTTTG -3′

R:5′-AATTCAAAAACAACAAGATGAAGAGCACCAACTCGAGTTGGTG
CTCTTCATCTTGTTG-3′

**RNA isolation and qRT-PCR**. Total RNA was extracted using TRIzol Reagent (Invitrogen) and reverse-transcribed using a High Capacity RNA-to-cDNA Kit (Applied Biosystems) under conditions provided by the supplier. Complementary DNA was quantified by real-time RT-PCR using ABI ViiA 7 System (Applied Biosystems). qRT-PCR was performed using SYBR Green reagents (Applied Biosystems) according to the manufacturer's instructions. GAPDH (glyceraldehyde 3-phosphate dehydrogenase) was used as an internal control. Primers used for qRT-RCR are listed below:

BRD9:F: 5′-GGCAAGATGGGCTATCTGAAG-3′
R: 5′-GGGAGTAGCTTACTGGAGAGC-3′
GAPDH:F: 5′-ACACCATGGGGAAGGTGAAG-3′
R: 5′-AAGGGGTCATTGATGGCAAC-3′

**Cell growth assay**. Cells infected with BRD9 shRNAs and control shRNA were seeded in a 96-well plates, respectively. Cell numbers were estimated by MTT (3-(4,5-dimethylthiazol-2-yl)-2,5-diphenyltetrazolium bromide) assay, using the Cell Proliferation Kit (I) (Roche) following the manufacturer's instructions. The resulting colored solution was quantified using an ELx800 Absorbance Microplate Reader (BioTek) at 570 nm with a reference wavelength of 630 nm.

**Soft agar assay**. The bottom layer was prepared with a 0.8% agarose (Invitrogen) solution in culture medium in six-well plates and the gel was allowed to set for 20 min. Cells ($2.5 \times 10^3$) were resuspended in 0.4% top agarose solution (in culture medium) then carefully placed on top of the bottom agarose in the six-well plates. The plates were incubated at 37 °C with 5% $CO_2$ until colonies were formed. Cell culture medium then was changed one to two times per week. After 2–3 weeks, cell colonies were stained using crystal violet (Sigma) and counted under a microscope.

**Mouse xenograft model in vivo**. Six- to eight-week-old female nude mice (Jackson Laboratory) were used for the xenograft assays. MDA-MB-231 cells and OVCAR8 were trypsinized and suspended in phosphate-buffered saline (PBS), then a total volume of 0.1 ml PBS containing MDA-MB-231 cells ($1.5 \times 10^6$) or OVCAR8 cells ($2 \times 10^6$) were injected subcutaneously into the mouse flank. Approximately 8 days later, tumors were detectable and tumor size was measured using a Vernier caliper. Tumor volumes were calculated by the formula $V = (4/3) \pi \times r^3$, where $r$ is the radius of a tumor. Tumor weights were measured at the end point of the experiment.

**Statistical analysis**. Statistical analysis was performed using R software. All results were expressed as mean ± SD and $p < 0.05$ indicated significance.

**Reporting summary**. Further information on experimental design is available in the Nature Research Reporting Summary linked to this article.

## Data availability

This study is based on the genomic profiling data generated by the TCGA project supported by the NCI and NHGRI. Information about TCGA and the TCGA research network can be found at the TCGA project website (http://cancergenome.nih.gov). The raw profiling data used for the current study are public available through the Genomic Data Commons (GDC) portal (https://gdc-portal.nci.nih.gov), the TCGA data portal (https://tcga-data.nci.nih.gov/tcga/), the GDAC Firehose of the Broad Institute (http://gdac.broadinstitute.org/), the TCGA Multi-Center Mutation Calling in Multiple Cancers (MC3) project (https://doi.org/10.7303/syn7214402), and TumorFusions data portal (http://tumorfusions.org/). The data generated by this study are public available through the Functional Cancer Genome data portal (FCG data portal, http://52.25.87.215/home/).

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

## Acknowledgements

We thank the TCGA project team. This work was supported, in whole or in part, the US National Institutes of Health (R01CA225929 to L.Z., R01CA142776 to L.Z., R01CA190415 to L.Z., P50CA083638 to L.Z., P50CA174523 to L.Z., and R01NS094533 to Y.F.), the Breast Cancer Alliance (L.Z.), the Marsha Rivkin Center for Ovarian Cancer Research (L.Z.), the Harry Fields Professorship (L.Z.), the Kaleidoscope of Hope Ovarian Cancer Foundation (L.Z.), the Ovarian Cancer Research Fund Alliance (X.H.), and the Foundation for Women's Cancer (X.H. and Y.Z.).

## Author contributions

Z.H., X.H., and L.Z. conceived and designed the research. Z.H., J.Y., X.W., and X.H. performed the computational analysis and statistical computations. J.Z., J.J., Y.Z., Y.W., Y.P., T.Z., X.Z., M.L., and X.H. performed the experiments. Z.H., K.T.M., J.L.T., Y.F., T.L. W., I.M.S., X.H., and L.Z. performed data collection, analysis, and general discussion. Z. H., N.L., X.H., and L.Z. wrote the paper.

## Additional information

**Competing interests:** The authors declare no competing interests.

