## [Peer Review File · Nature Communications]

Reviewers' comments:

Reviewer #1 (Remarks to the Author):

This manuscript by Hu et al. used sophisticated bioinformatics approaches to systematically examine copy number alterations, point mutations, and transcript fusions of a class of 73 histone acetylation modulator proteins (HAMPs) over 33 cancer types available in TCGA dataset. The manuscript is very well written and easily understandable. The presentation and visualization of the data and findings are impressive. The manuscript identified various HAMP alterations in different cancer types that could be interesting therapeutic targets for further characterization. As a proof-of-concept, the authors characterized BRD9, which is frequently amplified in multiple cancers, and demonstrated its role in increasing cell growth and inducing xenograft tumor growth in breast and ovarian cancer models. The manuscript will provide valuable resource for future studies of HAMPs in cancer. Some minor suggestions that could further improve the manuscript:

1. The manuscript is a bit descriptive until the last figure on BRD9, which however was described very hastily. It would be nice if BRD9 was emphasized as an example early on in bioinformatics analysis of copy number alterations, point mutations, and transcript fusions etc. Some more details on BRD9 gene alterations in different cancers would be appreciated and pave the way to detailed characterization in figure 8.
2. This study highlights the importance of HAMPs, in particular gain of histone acetylation and loss of HDAC in multiple cancer types. This should be discussed in the context of HDAC inhibitors and its failure in treating cancer.

Reviewer #2 (Remarks to the Author):

In the paper "Integrated genomic characterization of the genes encoding histone acetylation modulator proteins identifies potential therapeutic targets for precision epigenetic treatment of cancer," the authors present a comprehensive analysis of genetic alterations to pinpoint the precise 'targets' for cancer treatment. They perform SCNA, mutation, and fusion analysis to select HAMPs that may exhibit therapeutic potential. Their work with experimental validation is likely to be a useful tool for future studies. This manuscript is well written and data clearly presented. It is expected that this study should have an impact on this exciting research field. Comments are list as following.

Comments:

1. The levels of gene expression are different among different tissues. In Fig 1, the authors find that mRNAs of HAMPs were ubiquitously expressed across the 33 cancer types. However, information regarding the levels of mRNA compared to the corresponding normal tissue types is lacking. This statement could be further clarified with an additional heatmap of gene expression compared with their corresponding tissue/cell types.
2. In Fig 2, the authors identified 54 HAMPs as putative cancer-causing genes driven by SCNAs across cancers. However, the peak regions where somatic copy number alterations located may contain multiple genes at the significantly recurrent focal SNCA loci. Some of the genes may be oncogenes or tumor suppressor genes or just the passenger genes. It would be useful to list/analyze the co-altered genes within those 54 HAMPs.
3. To reduce false-negative rates of identifying driver gene mutations, the authors carefully integrate five methods to pool the genes that contain somatic mutations with selective pressures that cause tumor cell evolution. However, genes with mutations that promote tumorigenesis are not necessarily mutated with high frequencies. The list of genes could be strengthened by

analyzing with time status of each mutation. Early (trunk) mutation events will be likely to be driver gene mutations.

4. In Fig 5, the authors performed domain mutation and hotspot analysis to look for mutations within related functional domains of HAMPs. What are the HAMPs that contain those mutations? Do they have a high recurrent score?

5. Each mutation resulted from somatic SCNAs, mutations or fusions may influence tumor cells under different pathway/mechanisms and contribute to distinct parts of tumorigenesis. In Fig 7, the authors use the recurrent score of HAMPs to estimate cancer-driving potential for putative therapeutic targets. The authors should elaborate on the discussion section as to how to appropriately apply the recurrent score to find suitable targets.

6. In the Fig 1B, the authors indicate mRNA expression of HAMPs labeled with gray color are cancer type-specific. Explain what it means by "cancer type-specific." It is not clear how the selection process was made.

7. The authors should add an explanation in the legend of Fig 2C, 4C, and 7B on what it means with different color coding of HAMPs in the phylogenetic trees.

We would like to express our sincere gratitude to the reviewers for the enthusiasm they expressed towards the novelty and the significance of our work: “*The manuscript is very well written and easily understandable. The presentation and visualization of the data and findings are impressive The manuscript will provide valuable resource for future studies of HAMPs in cancer.*” [Reviewer #1]; “*This manuscript is well written and data clearly presented..... It is expected that this study should have an impact on this exciting research field.” [Reviewer #2].*

We are truly grateful to the editor and the reviewers for providing constructive criticism and thoughtful comments, which have helped us to significantly improve this manuscript. In response, we have **performed additional analyses and have considerably revised the text, figures and online supporting materials to address the issues that were raised.** Due to space limitation, we could not include all the new results in the main figures, and had to incorporate many of them in supplemental figures and tables. As a result, **the revised manuscript has eight (8) main figures, seven (7) supplemental figures and twenty-four (24) supplemental tables.** We believe that the revised manuscript has been significantly strengthened and all reviewers’ questions have been addressed.

In the point-by-point response letter, we used:

- Figure R to indicate the figures referred to in this response letter;
- Figure S to indicate the online supplemental figures in the revised manuscript;
- Figure to indicate the main figures in the revised manuscript.

POINT BY POINT RESPONSE

Reviewer #1:

This manuscript by Hu et al. used sophisticated bioinformatics approaches to systematically examine copy number alterations, point mutations, and transcript fusions of a class of 73 histone acetylation modulator proteins (HAMPs) over 33 cancer types available in TCGA dataset. The manuscript is very well written and easily understandable. The presentation and visualization of the data and findings are impressive. The

manuscript identified various HAMP alterations in different cancer types that could be interesting therapeutic targets for further characterization. As a proof-of-concept, the authors characterized BRD9, which is frequently amplified in multiple cancers, and demonstrated its role in increasing cell growth and inducing xenograft tumor growth in breast and ovarian cancer models. The manuscript will provide valuable resource for future studies of HAMPs in cancer. Some minor suggestions that could further improve the manuscript:

1. The manuscript is a bit descriptive until the last figure on BRD9, which however was described very hastily. It would be nice if BRD9 was emphasized as an example early on in bioinformatics analysis of copy number alterations, point mutations, and transcript fusions etc. Some more details on BRD9 gene alterations in different cancers would be appreciated and pave the way to detailed characterization in figure 8.

We appreciate that the reviewer suggested that we provide more detailed information about BRD9 genomic alterations in different cancer types. Followed the reviewer's comment, we summarized the genomic alterations of BRD9 (including SCNA, mutation and transcript fusion) for each TCGA cancer type. These results were added into the revised version of the manuscript as a new supplementary figure (Figure S7/Figure R1). In addition, followed by the reviewer's suggestion, BRD9 was emphasized as an example in early bioinformatics analysis. The data were described in the main text and were also showed in Figure 3B (the first panel).

Figure R1 (new Figure S7). Genomic alterations of BRD9 gene across 33 cancer types

A. Overview of BRD9 SCNAs across 33 common adult cancers. The cancer type is color-coded and labeled on the top of the figure. The upper four panels show focal peak, G-score, the 90th percentile of mRNA expression (FPKM), and the correlation between mRNA expression and predicted copy number (-log[p-value]) for each TCGA cancer type. The middle panel shows the predicted BRD9 copy numbers of each patient across 33 cancer types. In each cancer type, the patients are ordered by predicted BRD9 copy numbers. The cancer types with BRD9 recurrent amplification are marked as yellow. The lower two panels show the GISTIC status and WGD status of each patient. **B.** Frequency of BRD9 mutations across 33 common adult cancers. Note: in most cancer types, BRD9 gene has low mutation frequency (average 0.74% across all TCGA specimens). Only in UCEC and SKCM (both are high mutation burden cancers), BRD9 gene has relatively high mutation frequencies

(>2%). **C.** The lollipop plot illustrates the distribution and categories of somatic mutations in the BRD9 gene-coding sequences across all cancer types. The mutations are randomly distributed along the entire coding sequence, and no hotspot mutation was identified. **D.** Numbers of BRD9 transcript fusions across 33 common adult cancers. Note: only five transcript fusion events were identified in three cancer types (ESCA, KIRP, and LUSC). **E.** The fusions of BRD9 in TCGA samples are shown in Circos plots. The fused genes are illustrated as lines that connect two parental genes.

2. This study highlights the importance of HAMPs, in particular gain of histone acetylation and loss of HDAC in multiple cancer types. This should be discussed in the context of HDAC inhibitors and its failure in treating cancer.

We thank the reviewer for this important comment, i.e., the potential influence of HAMP copy number alteration on HDACi treatment responses should be carefully discussed. Followed the reviewer's suggestion, we have added additional discussion about this issue in the Discussion Section of the revised version of the manuscript. We appreciate this valuable and thoughtful suggestion.

Related discussion the Discussion Section:

“Notably, several HDAC genes, such as HDAC4 and HDAC10, showed significant, focal copy number loss, which is consistent with the experimental evidence that certain HDAC genes may serve as tumor suppressors during tumorigenesis. Furthermore, genetic depletion of HDACs in tumor cells leads to cell cycle arrest, apoptosis, and senescence, suggesting that HDACs are required for the survival and growth of tumor cells¹⁻³. Collectively, our findings indicate that targeting select HDAC isoforms in certain cancer types may be a more precise approach for the treatment of cancer compared to the use of pan-HDACis.”

“Among the three groups of HAMPs, HDACs are the best-characterized gene family and their inhibitors (HDACis) have been approved to treat hematopoietic malignancies^{1, 2, 4-8}. Although HDACis have also been widely tested in solid tumors in early clinical phases, the results from the clinical trials have revealed limited anti-cancer activity. Our results indicate that certain HDAC genes (e.g., HDAC4 and HDAC10) focally lose copy numbers in selected cancer types. The copy numbers lost in tumors may depend on the remaining HDAC allele; thus, further suppression of HDACs may lead to greater anti-cancer effects in these tumors. In addition,

given that different HDAC genes have distinct genomic alterations in cancer, the development of isoform-selective HDACis is urgently needed for future clinical application.”

Reviewer #2:

In the paper "Integrated genomic characterization of the genes encoding histone acetylation modulator proteins identifies potential therapeutic targets for precision epigenetic treatment of cancer," the authors present a comprehensive analysis of genetic alterations to pinpoint the precise ‘targets’ for cancer treatment. They perform SCNA, mutation, and fusion analysis to select HAMPs that may exhibit therapeutic potential. Their work with experimental validation is likely to be a useful tool for future studies. This manuscript is well written and data clearly presented. It is expected that this study should have an impact on this exciting research field. Comments are list as following.

1. The levels of gene expression are different among different tissues. In Fig 1, the authors find that mRNAs of HAMPs were ubiquitously expressed across the 33 cancer types. However, information regarding the levels of mRNA compared to the corresponding normal tissue types is lacking. This statement could be further clarified with an additional heatmap of gene expression compared with their corresponding tissue/cell types.

We appreciate that the reviewer suggested that we add an additional heatmap for gene expression in corresponding normal adjacent tissues. Followed the reviewer’s comment, we generated a heatmap to present the mRNA expression levels of HAMPs in the TCGA corresponding normal adjacent tissues from 18 cancer types in which sufficient RNA-seq data from normal tissues are available ($n \geq 5$) (Figure R2A). In addition, we also analyzed a large collection of RNA-seq data for 2,012 cancer cell lines, and summarized the results as a heatmap (Figure R2C). Consistence with their expression in primary tumor specimens, we found that most HAMPs were ubiquitously expressed across normal tissues or cancer cell lines. Only few HAMPs, such as BRDT, were expressed in a lineage-specific manner. Although these lineage-specific HAMP genes were mainly detected in the cancer types derived from the tissues in which the corresponding HAMP genes are normally expressed, they were also ectopically expressed in a small fraction of other cancer types. For example, the testis-specific BRD gene BRDT was not solely detected in testicular germ cell tumors (TGCT); it was also found in a small fraction of lung cancers (25.34% of lung adenocarcinomas [LUAD] and 16.97% of lung squamous cell carcinoma [LUSC]), uterine carcinosarcoma (UCS;16.07%), and esophageal carcinoma (ESCA;

11.18%). This finding indicates the therapeutic potential of targeting lineage-specific HAMPs in certain cancer types. These results were added into the revised version of the manuscript as a new supplementary figure (Figure S1/Figure R2).

Figure R2 (new Figure S1). Ubiquitous expression of the HAMP genes across normal tissues and cancers
A. Heatmap shows the mRNA expression levels of the HAMP genes across corresponding adjacent normal tissues of TCGA (n=715 from 18 primary sites). The intensity of purple indicates the percentile (25th, 50th, 75th,

and 90th) of the FPKM value of each HAMP in a given tissue type. **B.** Heatmap shows the mRNA expression levels of the HAMP genes across TCGA tumors (n=10,201, with samples of 33 cancer types from 27 primary sites). The intensity of purple indicates the percentile (25th, 50th, 75th, and 90th) of the FPKM value of each HAMP in a given cancer type. **C.** Heatmap shows the mRNA expression levels of the HAMP genes across large-collection of human cancer cell lines (n=2,012, with samples of 30 cancer types). The intensity of purple indicates the percentile (25th, 50th, 75th, and 90th) of the FPKM value of each HAMP in a given cancer type. The phylogenetic trees were generated by multiple sequence alignments of the full-length sequences of the proteins.

2. In Fig 2, the authors identified 54 HAMPs as putative cancer-causing genes driven by SCNAs across cancers. However, the peak regions where somatic copy number alterations located may contain multiple genes at the significantly recurrent focal SNCA loci. Some of the genes may be oncogenes or tumor suppressor genes or just the passenger genes. It would be useful to list/analyze the co-altered genes within those 54 HAMPs.

We thank the reviewer for raising this important question. We agree with the reviewer that most peak regions which were identified by GISTIC contain multiple genes, and a list of co-altered genes will be useful information for this study. Followed by the reviewer's comments, we mapped each focally altered region containing HAMP to the human genome, and identified the co-altered genes within each locus. This result was added into the revised version of the manuscript as Table S7.

3. To reduce false-negative rates of identifying driver gene mutations, the authors carefully integrate five methods to pool the genes that contain somatic mutations with selective pressures that cause tumor cell evolution. However, genes with mutations that promote tumorigenesis are not necessarily mutated with high frequencies. The list of genes could be strengthened by analyzing with time status of each mutation. Early (trunk) mutation events will be likely to be driver gene mutations.

We appreciate that the reviewer suggested that we analyze the timing of the mutational processes of each mutation. Followed the reviewer's comment, we analyzed the timing of the mutational processes using the ABSOLUTE algorithm⁹. This result was presented in the revised version of the manuscript as Table S15.

4. In Fig 5, the authors performed domain mutation and hotspot analysis to look for mutations within related functional domains of HAMPs. What are the HAMPs that contain those mutations? Do they have a high recurrent score?

We thank the reviewer for raising this important question. Followed the reviewer's comments, we analyzed the HAMPs that contain the mutated domains in each cancer type. Many of HAMPs containing those mutations have a high recurrent mutation score, such as EP300 and CREBBP, which contain HAT-KAT11 domain. These results were added into the revised version of the manuscript as Table S17.

5. Each mutation resulted from somatic SCNAs, mutations or fusions may influence tumor cells under different pathway/mechanisms and contribute to distinct parts of tumorigenesis. In Fig 7, the authors use the recurrent score of HAMPs to estimate cancer-driving potential for putative therapeutic targets. The authors should elaborate on the discussion section as to how to appropriately apply the recurrent score to find suitable targets.

We thank the reviewer for this important comment. i.e., how to appropriately apply the recurrent score to find suitable targets for cancer treatment should be carefully discussed. Followed the reviewer's suggestion, we have added additional discussion about this issue in the Discussion Section of the revised version of the manuscript. We appreciate this valuable and thoughtful suggestion.

Related discussion in the Discussion Section:

“In our study, an overall recurrent score for each HAMP was estimated at a pan-cancer level by an unweighted numeric sum of the numbers of recurrent events. This overall recurrent score can prioritize the potential candidates for future anticancer drug development. A HAMP with a high score indicates its recurrent alterations are common genomic events across multiple adult cancer types. Thus, successful strategies targeting such HAMPs may benefit relatively larger proportions of patients from different cancer types. For example, BRD9 shows highest overall recurrent score among HAMPs (e.g., focally amplified in nine cancer types), strongly suggesting that targeting BRD9 by small molecular inhibitors may be a novel treatment strategy with wide clinical application for cancer therapy. Finally, given that many HAMPs with high scores are understudied genes (e.g., BRD9), the overall recurrent score may also be used to prioritize the genes for basic biological studies in context of cancer.”

6. In the Fig 1B, the authors indicate mRNA expression of HAMPs labeled with gray color are cancer type-specific. Explain what it means by “cancer type-specific.” It is not clear how the selection process was made.

We thank the reviewer for raising this question. Cancer type-specific expression of a given HAMP gene means its mRNA expression was not detectable in all cancer types. I.e., we defined the HAMPs which were not expressed in all cancer types as cancer type-specific HAMPs. As showed in Figure 1A and 1B, most of HAMPs (65/73; 89%) were ubiquitously expressed in all 33 cancer types. However, some HAMPs are specifically expressed in certain cancer types. For example, BRDT was detected in testicular germ cell tumors (TGCT) as well as in a small fraction of lung cancers (25.34% of lung adenocarcinomas [LUAD] and 16.97% of lung squamous cell carcinoma [LUSC]), uterine carcinosarcoma (UCS; 16.07%), and esophageal carcinoma (ESCA; 11.18%). These kinds of HAMPs were defined as “cancer type-specific” in this study. Here, positive expression of a HAMP in a given cancer type was defined as its mRNA expression was reliably detected in at least 10% of tumor specimens (i.e., the 90th percentile of fragments per kilobase of transcript per million mapped reads [FPKM] value ≥ 1).

7. The authors should add an explanation in the legend of Fig 2C, 4C, and 7B on what it means with different color coding of HAMPs in the phylogenetic trees.

We thank the reviewer for raising this question. The color coding for Figure 2C, 4C and 7B has been added into the corresponding figure legends in the revised version of the manuscript:

Figure 2C:

Black and gray indicate a gene with an overall G-score ≥ 0.8 and < 0.8 , respectively.

Figure 4C:

Black and gray indicate a gene with an overall M-score ≥ 0.4 and < 0.4 , respectively.

Figure 6B:

Black and gray indicate a gene with fusion events ≥ 10 and < 10 , respectively.

Figure 7B:

Black and gray indicate a gene with an overall recurrent score ≥ 6 and < 6 , respectively.

References

1. West, A.C. & Johnstone, R.W. New and emerging HDAC inhibitors for cancer treatment. *J Clin Invest* **124**, 30-39 (2014).
2. Ceccacci, E. & Minucci, S. Inhibition of histone deacetylases in cancer therapy: lessons from leukaemia. *Br J Cancer* **114**, 605-611 (2016).
3. Seto, E. & Yoshida, M. Erasers of histone acetylation: the histone deacetylase enzymes. *Cold Spring Harb Perspect Biol* **6**, a018713 (2014).
4. Jones, P.A., Issa, J.P. & Baylin, S. Targeting the cancer epigenome for therapy. *Nat Rev Genet* **17**, 630-641 (2016).
5. Ribich, S., Harvey, D. & Copeland, R.A. Drug Discovery and Chemical Biology of Cancer Epigenetics. *Cell Chem Biol* **24**, 1120-1147 (2017).
6. Shortt, J., Ott, C.J., Johnstone, R.W. & Bradner, J.E. A chemical probe toolbox for dissecting the cancer epigenome. *Nat Rev Cancer* **17**, 160-183 (2017).
7. Cai, S.F., Chen, C.W. & Armstrong, S.A. Drugging Chromatin in Cancer: Recent Advances and Novel Approaches. *Mol Cell* **60**, 561-570 (2015).
8. Morgan, M.A. & Shilatifard, A. Chromatin signatures of cancer. *Genes Dev* **29**, 238-249 (2015).
9. Carter, S.L. *et al.* Absolute quantification of somatic DNA alterations in human cancer. *Nature biotechnology* **30**, 413-421 (2012).
1. West, A.C. & Johnstone, R.W. New and emerging HDAC inhibitors for cancer treatment. *J Clin Invest* **124**, 30-39 (2014).
2. Ceccacci, E. & Minucci, S. Inhibition of histone deacetylases in cancer therapy: lessons from leukaemia. *Br J Cancer* **114**, 605-611 (2016).
3. Seto, E. & Yoshida, M. Erasers of histone acetylation: the histone deacetylase enzymes. *Cold Spring Harb Perspect Biol* **6**, a018713 (2014).
4. Jones, P.A., Issa, J.P. & Baylin, S. Targeting the cancer epigenome for therapy. *Nat Rev Genet* **17**, 630-641 (2016).
5. Ribich, S., Harvey, D. & Copeland, R.A. Drug Discovery and Chemical Biology of Cancer Epigenetics. *Cell Chem Biol* **24**, 1120-1147 (2017).
6. Shortt, J., Ott, C.J., Johnstone, R.W. & Bradner, J.E. A chemical probe toolbox for dissecting the cancer epigenome. *Nat Rev Cancer* **17**, 160-183 (2017).

7. Cai, S.F., Chen, C.W. & Armstrong, S.A. Drugging Chromatin in Cancer: Recent Advances and Novel Approaches. *Mol Cell* **60**, 561-570 (2015).
8. Morgan, M.A. & Shilatifard, A. Chromatin signatures of cancer. *Genes Dev* **29**, 238-249 (2015).
1. West, A.C. & Johnstone, R.W. New and emerging HDAC inhibitors for cancer treatment. *J Clin Invest* **124**, 30-39 (2014).
2. Ceccacci, E. & Minucci, S. Inhibition of histone deacetylases in cancer therapy: lessons from leukaemia. *Br J Cancer* **114**, 605-611 (2016).
3. Seto, E. & Yoshida, M. Erasers of histone acetylation: the histone deacetylase enzymes. *Cold Spring Harb Perspect Biol* **6**, a018713 (2014).

REVIEWERS' COMMENTS:

Reviewer #1 (Remarks to the Author):

The revision has satisfactorily addressed my critiques.